# Learning and Processing the Ordinal Information of Temporal Sequences in Recurrent Neural Circuits

**Xiaolong Zou**[1,4,5,*]
zouxiaolong@qiyuanlab.com

**Zhikun Chu**[2,*]
chukunzhi@outlook.com

**Qinghai Guo**[7]
guoqinghai@huawei.com

**Jie Cheng**[7]
jiecheng2009@gmail.com

**Bo Hong**[1,6]
hongbo@tsinghua.edu.cn

**Si Wu**[4,5]
siwu@pku.edu.cn

**Yuanyuan Mi**[3,⊤]
miyuanyuan@tsinghua.edu.cn

1, Qiyuan Lab, Beijing, China.
2, Center for Neurointelligence, School of Medicine, Chongqing University.
3, Department of Psychology, Tsinghua University
4, School of Psychological and Cognitive Sciences,
Beijing Key Laboratory of Behavior and Mental Health,
IDG/McGovern Institute for Brain Research,
Peking-Tsinghua Center for Life Sciences, Center of Quantitative Biology,
Academy for Advanced Interdisciplinary Studies, Peking University, Beijing, China
5, Beijing Academy of Artificial Intelligence, Beijing,China.
6, Biomedical Engineering,School of Medicine, Tsinghua University
7, Huawei Technologies.
*: Equal contributions. ⊤ : Corresponding authors.

## Abstract

Temporal sequence processing is fundamental in brain cognitive functions. Experimental data has indicated that the representations of ordinal information and contents of temporal sequences are disentangled in the brain, but the neural mechanism underlying this disentanglement remains largely unclear. Here, we investigate how recurrent neural circuits learn to represent the abstract order structure of temporal sequences, and how the disentangled representation of order structure facilitates the processing of temporal sequences. We show that with an appropriate training protocol, a recurrent neural circuit can learn tree-structured attractor dynamics to encode the corresponding tree-structured orders of temporal sequences. This abstract temporal order template can then be bound with different contents, allowing for flexible and robust temporal sequence processing. Using a transfer learning task, we demonstrate that the reuse of a temporal order template facilitates the acquisition of new temporal sequences, if these sequences share the same or partial ordinal structure. Using a key-word spotting task, we demonstrate that the tree-structured attractor dynamics improves the robustness of temporal sequence discrimination, if the ordinal information is the key to differentiate these sequences. We hope that this study gives us insights into the mechanism of representing the ordinal information of temporal sequences in the brain, and helps us to develop brain-inspired temporal sequence processing algorithms.

37th Conference on Neural Information Processing Systems (NeurIPS 2023).

# 1 Introduction

A temporal sequence involves a set of items or events that unfold in a specific order over time. Temporal sequence processing is fundamental in many cognitive functions [1], such as speech recognition, language comprehension, motor control, and memory formation. Effective temporal sequence processing relies on extracting the temporal structure of a sequence, which includes particularly the ordinal information, i.e., the order of events' occurrence. A large volume of studies has suggested that the representations of ordinal information and contents of a temporal sequence are disentangled in the brain [1–6]. For instances, studies using single-unit recording in primates have found that a significant proportion of neurons in the dorsolateral prefrontal and intraparietal cortexes are sensitive to the temporal order of presented visual items, but not to their physical attributes [5], and similar temporal order representations were also observed in the auditory and linguistic sequence processing [2]. The imaging study has shown that the human brain can encode the phonetic order during speech processing, regardless of the phonetic contents [2]. Recent large-scale neuron recordings have revealed that the temporal order structure can be robustly encoded in a low-dimensional space formed by coordinated neuron population activities [4]. The advantage of disentangled representation of order structure is that it enables the brain to flexibly and rapidly process temporal sequences via conjunctive coding [1], i.e, the brain can easily generate various temporal sequences by combining the temporal order with different contents. This was evident in a finger sequence learning task [6], where the brain stored a temporal order template and reused it to combine actions to generate various finger action sequences. Understanding how ordinal information is acquired and stored in the brain is crucial for us to understand how temporal sequences are processed in cognitive functions.

Remarkably, the representations of temporal orders of related sequences display a tree-like structure [7]. For example, in speech recognition, spoken words are represented by temporal sequences formed by primitive syllables or phonemes. As illustrated in Fig.1A, the words "water" and "wash" share a common syllable "w-ao" at the first item and differ at the second syllable, and so do the words "your" and "year". Thus, the temporal orders of these four sequences form a two-layer tree: starting from the root, the words "water" and "wash" form a branch in the first layer as they share the same first syllable, and the words "your" and "year" form the other branch; in the second layer, the words "water" and "wash", and the words "your" and "year", are further branched due to their difference at the second syllable. Experimental data has indicated that such tree-structured representation of ordinal information is employed in the frontal cortex to support hierarchical motor control [8].

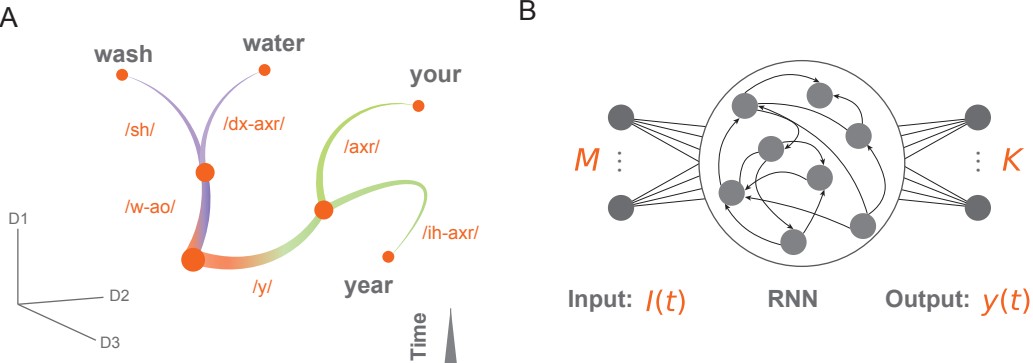

Figure 1: (A) A schematic diagram illustrating the tree-structured temporal orders of four spoken words. Words 'water' and 'wash' form a branch in the first layer because of the shared syllable 'w-ao', and they are branched in the second layer due to the different second syllables, 'dx-axr' and 'sh'. Similarly, words 'your' and 'year' form other branches. (B) A schematic diagram of our neural circuit model. It consists of three parts: an input layer of dimension $M$ conveying temporal sequences, a recurrent neural network (RNN) of $N$ neurons encoding the ordinal structure, and an output layer discriminating $K$ different temporal sequences.

Despite that the representation of ordinal information in the brain has been well recognized in the field, the neural mechanism of implementing this representation remains largely unclear [1, 7]. A number of computational models has been proposed to mimic how temporal sequences are stored

in neural circuits, and they typically consider that neurons are asymmetrically connected along the path of a sequence of network states representing the temporal sequence [10–12, 15]. Among the proposed models, the framework of heteroclinic channel is inspiring [10], which considers that a temporal sequence is stored as a sequence of saddle points connected by unstable paths in the state space of the network, forming the so-called a heteroclinic channel, and under the drive of external inputs, the network state evolves along the channel to activate stored items in the right time order. In most previous works [11, 13–15], the asymmetric neuronal connections accounting for storing temporal sequences are either designed by hand or constructed by applying an asymmetric Hebbian learning rule (note that the application of a Hebbian learning rule implicitly assumes that neural representations of sequential items are known, which does not really model the learning process), and they did not address how a neural circuit learns to represent the temporal order structure from data. A few studies explored how temporal sequences are learned in a recurrent neural network [15], but they only considered the simple chain-structured temporal order and did not study the more general tree-structured orders. The machine learning society also studied the learning of temporal sequences (such as speech) by using artificial neural networks, but they have not considered the representation of disentangled ordinal information [9].

In this work, we explore how biologically plausible recurrent neural circuits learn to represent the tree-structured orders of temporal sequences and how this disentangled representation of order structure facilitates the processing of temporal sequences. We show that with an appropriate training protocol, a recurrent neural circuit can learn a set of tree-structured attractor states to encode the corresponding tree-structured orders of temporal sequences, where each path from the root to a leaf represents a pattern of temporal order, with nodes on the path indicating the order hierarchy (see Fig.1A). This abstract order template can then be bound with different contents, allowing for flexible and robust temporal sequence processing. To demonstrate this advantage, we carry out two experiments. First, in a transfer learning task, we show that the reuse of the order template accelerates the acquisition of new temporal sequences, if these sequences share the same or partial ordinal structure. Second, in a key-word spotting task, we show that using the tree-structured attractor dynamics improves the robustness of temporal sequence discrimination, if the ordinal structure is the key to differentiate these sequences. We hope that this study facilitates our understanding of how temporal sequences are represented and processed in the brain, and helps us to develop brain-inspired algorithms for temporal sequence processing.

## 2 Learning the ordinal structure of temporal sequences

### 2.1 The neural circuit model

The model we consider consists of three parts (Fig.1B): an input layer which conveys the information of temporal sequences, a recurrent neural network (RNN) which stores the ordinal structure of temporal sequences, and an output layer which discriminates temporal sequences. Our idea is that by training the model to perform a temporal sequence discrimination task, the RNN learns to represent the ordinal structure of given temporal sequences in its dynamics. The details of the model are introduced below.

Denote $\mathbf{I}(t) = \{I_m(t)\}$, for $m = 1, \ldots, M$, with $M$ the dimension, the activity of the input layer conveying the temporal sequence. The input layer is connected to the RNN through a feedforward connection matrix $\mathbf{W}^{in} = \{W_{nm}^{in}\}$, for $m = 1, \ldots, M$ and $n = 1, \ldots, N$, with $N$ the number of neurons in the RNN. Neurons in the RNN are recurrently connected with each other with a matrix $\mathbf{W}^{rec} = \{W_{nj}^{rec}\}$, for $n, j = 1, \ldots, N$, and they also receive a background input $\mathbf{I}^b = \{I_n^b\}$, for $n = 1, \ldots, N$. Denote $x_n(t)$ the total input received by the $n$th neuron in the RNN at time $t$ and $r_n(t)$ the corresponding neuronal activity. The dynamics of $x_n(t)$ is written as,

$$\tau \frac{dx_n(t)}{dt} = -x_n(t) + \sum_{j=1}^{N} W_{nj}^{rec} r_j(t) + \sum_{m=1}^{M} W_{nm}^{in} I_m(t) + I_n^b, \tag{1}$$

where $\tau$ is the time constant. The nonlinear relationship between $x_n(t)$ and $r_n(t)$ is given by $r_n(t) = \tanh[x_n(t)]$.

The RNN is connected to the output layer through a feedforward matrix $\mathbf{W}^{out} = \{W_{kn}^{out}\}$, for $n = 1, \ldots, N$ and $k = 1, \ldots, K$, with $K$ the number of neurons in the output layer. $K$ is also the

number of temporal sequences, since each neuron reads out one class of sequences. Denote $y_k(t)$ the activity of the $k$th neuron in the output layer, whose dynamics is given by,

$$y_k(t) = \sum_{n=1}^{N} W_{kn}^{out} r_n(t). \qquad (2)$$

## 2.2 The training protocol

The details of the training protocol are presented in Supplementary Information (SI). Here, we highlight two settings which are critical for the RNN to extract the ordinal structure of temporal sequences.

Firstly, we augment training data by varying the duration between neighbouring items in a temporal sequence (see illustration in Fig.2A). Specifically, this is done by adding noisy signals of varied duration between neighbouring primitive items of the temporal sequence (see more details below). The underlying mechanism is intuitively understandable. The augmented data imposes that the model needs to realize invariant recognition with respect to the warping of a temporal sequence, which forces the RNN to extract the abstract ordinal structure of the temporal sequence. This is also biologically reasonable, as in speech recognition [2, 3], the varied speed of spoken words forces the brain to process temporal sequences relying mainly on their ordinal information.

Secondly, we set the target function, i.e., the target activity of a read-out neuron in response to the temporal sequence it encodes, to be continuous in time, and its value increases step by step as items in the temporal sequence unfold one by one (see illustration in Fig.2A). The underlying mechanism is intuitively understandable. Since not all temporal sequences are differentiable by early items, neuronal responses should be low initially, and their values increase along with the exposure of followed items to reflect that sequence recognition is a process of evidence accumulation over time. Interestingly, such ramping activity of read-out neurons agree with the experimental finding on decision-making neurons in monkeys [1–3].

After constructing input data and target functions, we apply the standard algorithm, back-propagation through time, to train the model, which optimizes the parameters $\left( \mathbf{W}^{in}, \mathbf{W}^{rec}, \mathbf{W}^{out}, \mathbf{I}^{bg} \right)$. For details, see Sec. A in SI.

## 2.3 A synthetic task and the model behavior

To demonstrate that our model works, we first apply it to a synthetic task of discriminating four temporal sequences ($K = 4$). These four temporal sequences are constructed by combining two of three primitive items with varied orders (like syllables forming spoken words). As illustrated in Fig.2A, the three primitive items are 3-dimensional ($M = 3$) continuous-time vectors within an interval $\Delta t$, denoted as $\mathbf{a} = [1, 1, 0]^T, \mathbf{b} = [1, 0, 1]^T$, and $\mathbf{c} = [0, 1, 1]^T$, respectively. The contents of four temporal sequences are expressed as $\mathbf{S}^k = \{S_1^k, S_2^k\}$, with $k$ the index of the sequence and $S_1^k, S_2^k$ the first and second items. Since in this study, the exact duration of a temporal sequence is not important, for clearance, we also use $\mathbf{S}^k$ to denote a sequence having the contents $\{S_1^k, S_2^k\}$ with unspecified duration. The four classes of temporal sequences are therefore expressed as, $\mathbf{S}^1 = \{\mathbf{a}, \mathbf{b}\}$, $\mathbf{S}^2 = \{\mathbf{a}, \mathbf{c}\}, \mathbf{S}^3 = \{\mathbf{b}, \mathbf{a}\}, \mathbf{S}^4 = \{\mathbf{b}, \mathbf{c}\}$, and their temporal orders form a two-layer tree structure as shown in Fig.2B. To augment the training data, we carry out two operations. Firstly, we randomly set the duration between neighboring primitive items to be $\Delta T$, which satisfies a uniform distribution in the range of $[0, \Delta T_{max}]$. Secondly, we add noises in the duration, which are given by $\eta_i(t) = \sigma \xi_i(t)$, for $i = 1, \cdots, M$ and $t \in (0, \Delta T)$, where $\xi_i(t)$ denotes Gaussian white noises of zero mean and unit variance, and $\sigma$ the noise strength.

After training, we observe that the model learns to discriminate four temporal sequences in a coarse-to-fine manner. An example of discriminating the temporal sequence $\mathbf{S}^1$ is presented in Fig.2C. Initially, both activities of neurons 1 (red) and 2 (green) increase, since $\mathbf{S}^1$ and $\mathbf{S}^2$ are not differentiable by their first item, i.e., $S_1^1 = S_1^2$; while neurons 3 (orange) and 4 (blue) keep silent. When the second item appears, only the activity of neuron 1 keeps increasing, while the activity of neuron 2 drops, indicating that the model recognizes $\mathbf{S}^1$. The above results can be generalized to more than four temporal sequences, see Sec. A in SI for $K = 8, 12$.

## 2.4 Visualizing the learned tree-structured attractor dynamics

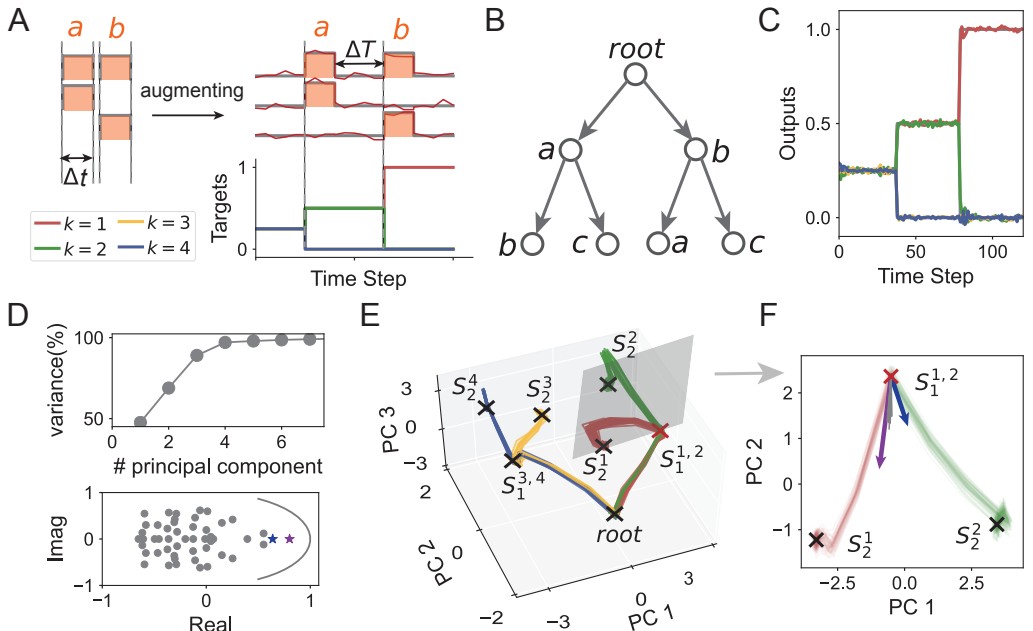

Figure 2: Unveiling the performance and the mechanism of the model using synthetic data. (A) Illustrating the constructions of temporal sequences and target functions. Two primitive items, $(\mathbf{a}, \mathbf{b})$, are combined to form the temporal sequence $\mathbf{S}^1$, with an interval $\Delta T$ inserted between two items, where $\Delta T$ is sampled from a uniform distribution in the range of $[0, \Delta T_{max}]$. Noises are added in the interval. Along with the unfold of two items in $\mathbf{S}^1$, the target function of read-out neuron 1 (red) increases step by step, the target function of read-out neuron 2 (green) first increases and then decreases, and the target functions of read-out neurons 3 (orange) and 4 (blue) always keep silent. (B) The orders of four temporal sequences $(\mathbf{S}^1, \mathbf{S}^2, \mathbf{S}^3, \mathbf{S}^4)$ form a two-layer tree. (C) An example of the model performance when recognizing the temporal sequence $\mathbf{S}^1$. It operates in a coarse-to-fine manner. When the first item appears, both the activities of read-out neurons 1 and 2 increase, since $\mathbf{S}^1$ and $\mathbf{S}^1$ are not differentiable at this stage; when the second item appears, only the activity of read-out neuron 1 keeps increasing, indicating the model accomplishes the discrimination task. (D) (upper panel) PCA analysis shows that the top 3 PCs explain nearly $90\%$ of the variance of neuronal responses in the RNN. (lower panel) The eigenvalue distribution of the attractor state $S_1^{1,2}$ in Fig.2E. (E) The visualization of the tree-structured neural response trajectories to four temporal sequences, $(\mathbf{S}^1, \mathbf{S}^2, \mathbf{S}^3, \mathbf{S}^4)$, in the reduced space spanned by 3 PCs. Each item in a temporal sequence is stored as an attractor in the RNN (marked by cross). (F) The transition probabilities from node $S_1^{1,2}$ to other states, which are quantified by the eigenvalues of the corresponding Jacobian matrix. The two eigenvectors having the largest two eigenvalues (reflected by their lengths) point to nodes $S_2^1$ and $S_2^2$. The parameters used are given in Sec. F in SI.

We carry out intensive analysis to visualize what have been learned in the dynamics of RNN.

### 2.4.1 Tree-structured representation trajectories

To visualize the high-dimensional dynamics of the RNN ($N = 50$), we apply principle component analysis (PCA) to reduce its dimensionality. PCA is done by using the activities of RNN neurons in response to 1000 testing temporal sequences (for details, see Sec. B.1 in SI). It turns out the top 3 PCs explain nearly $90\%$ of the variance of neuronal responses (Fig.2D, upper panel), indicating that after training, the RNN performs computation in a low-dimensional subspace. By projecting RNN activities onto the top 3 PCs, we obtain the trajectories of the network state in responses to the four temporal sequences (averaged over 25 trials), and observe that they display a tree structure, agreeing with the ordinal structure of the learned four temporal sequences (Fig.2E, trajectories for different sequences are marked by different colours).

### 2.4.2 Tree-structured attractor states

We continue to analyze the state space of the RNN, and find that it has totally 7 stable points (attractors), which are the nodes of the tree: the root, branching points, and end points (marked by crosses in Fig.2E), indicating that primitive items are stored as attractors in the RNN. The detailed analysis is presented in Sec. B.2 in SI. Here, we sketch the main idea. We first use an optimization method to find out all fixed points of the RNN dynamics [4], which turn out to coincide with the 7 nodes of the tree. We then analyze the stability of each node. An example of stability analysis of node $S_1^{1,2}$ is depicted in Fig.2D (lower panel), which shows that the real parts of all eigenvalues of the Jacobian matrix are small than 1, indicating that the node is stable with respect to noise perturbation.

### 2.4.3 Tree-structured attractor dynamics

We further analyze the transition dynamics of attractor states (nodes). The detailed analysis is presented in Sec. B.3 in SI. Here, we sketch the main idea. Consider node $S_1^{1,2}$ in the first layer of the tree (Fig.2E). We linearize the RNN dynamics around the node and compute all eigenvectors of the Jacobian matrix. The larger the eigenvalue is, the more unstable the network state along the eigenvector under noise perturbation. As shown in Fig.2F, the most two unstable directions (the length of eigenvector reflecting the eigenvalue) point to nodes $S_2^1$ and $S_2^2$ in the second layer, respectively. This implies that starting from the node $S_1^{1,2}$, the network state can travel to the node $S_2^1$ or $S_2^2$ depending on the second item, but has difficulty to travel to other nodes.

In summary, the above results reveal that after training, the tree-structured attractor dynamics emerges in the RNN, which encode the ordinal structure of the learned temporal sequences. Upon receiving a temporal sequence, the network state evolves from the root to the leaf, node by node, following the order hierarchy of the temporal sequence. Notably, unlike the framework of heteroclinic channel, our model stores primitive items as stable attractors rather than saddle points, and the transition between two nodes requires the followed item to be the input drive. As demonstrated below, this attractor dynamics enables our model to discriminate temporal sequences robustly against temporal warping.

## 3 Processing temporal sequences with the ordinal structure

### 3.1 Disentangled representation of ordinal structure facilitates transfer learning

The key advantage of disentangled representation of order structure is that it allows flexible generation to new temporal sequences via conjunctive code, that is, the neural system can combine the stored order template with different contents to form different temporal sequences. In our model, the RNN learns to extract the abstract ordinal structure of temporal sequences in the form of tree-structured attractor dynamics. After training, the association between the RNN and the training data is released, and the RNN can be reused to encode new temporal sequences having the same or partial temporal structure. Mathematically, this can be done by freezing the recurrent connections in the RNN (and hence the tree-structured attractor dynamics), while only learning the feedforward connections from the input layer to the RNN and from the RNN to the output layer, when a new temporal sequence processing task arises. In such a way, it is expected that the pre-trained RNN helps the model to speed up the acquisition of new temporal sequences having the same or partial ordinal structure, achieving the so-called transfer learning.

To demonstrate the above idea, we design a transfer learning task. We consider that the RNN has extracted the ordinal structure of the synthetic data described in Sec.2.3, and we now apply the model to solve a new task of discriminating new four temporal sequences. These four temporal sequences are composed by three primitive phonemes chosen from the TIMIT dataset, which are "pcl", "tcl", and "pau" (for the details of sequence construction, see Sec. C in SI), and the four sequences are written as "pcl-tcl", "pcl-pau", "tcl-pcl" and "tcl-pau", which have the same order structure as the synthetic data (Fig.3A-B). Following the same procedure as described in Sec.2.2, we augment the training data by inserting time intervals between neighboring items, and choose their ramping target functions accordingly. During transfer learning, we freeze the recurrent connections in the RNN and only optimize the feedforward connections. As comparison, we also train the model without freezing the RNN. The results are shown in Fig.3A-B. We observe that: 1) after transfer learning, the RNN retains the tree-structured representation trajectories for the learned temporal sequences, indicating that the model adjusts the feedforward connections to match the tree-structured attractors of the

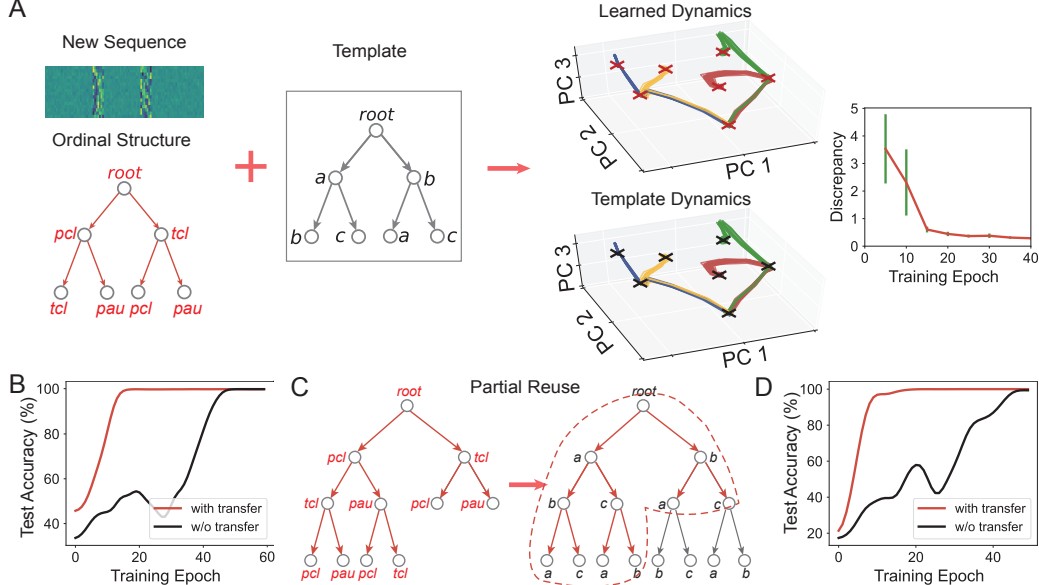

Figure 3: Disentangled representation of ordinal structure facilitates transfer learning. (A-B) Illustrating a transfer learning task, in which new temporal sequences have the same ordinal structure as the template. (A) From left to right. Four new sequences are constructed by using three primitive items (phonemes), and they have the same tree-structured orders as the template stored in the RNN. By freezing the recurrent connections in the RNN, the model learns the same tree-structured representation trajectories as the old ones, and their discrepancy is negligible after training (for the measurement of the discrepancy, see Sec. C in SI). (B) The learning of the model is accelerated significantly compared to the case of no transfer learning. (C) A schematic diagram of a transfer learning task, in which the ordinal structure of new temporal sequences only covers part of the template. (D) The learning of the model in (C) is accelerated significantly compared to the case of no transfer learning. For the details of model parameters and performances, see Sec. C, F in SI.

RNN (note that freezing recurrent connections in the RNN does not ensure that the representation trajectories are also fixed); 2) the speed of transfer learning is accelerated significantly compared to that without freezing the RNN.

We further consider a transfer learning task, in which the ordinal structure of new temporal sequences is only part of the order template stored in the RNN (more details see Sec. C in SI). Firstly, we create a task of discriminating eight sequences, which are constructed by the synthetic primitive items given in Sec.2.3. They are: $\mathbf{S}^1 = \{\mathbf{a}, \mathbf{b}, \mathbf{a}\}$, $\mathbf{S}^2 = \{\mathbf{a}, \mathbf{b}, \mathbf{c}\}$, $\mathbf{S}^3 = \{\mathbf{a}, \mathbf{c}, \mathbf{a}\}$, $\mathbf{S}^4 = \{\mathbf{a}, \mathbf{c}, \mathbf{b}\}$, $\mathbf{S}^5 = \{\mathbf{b}, \mathbf{a}, \mathbf{b}\}$, $\mathbf{S}^6 = \{\mathbf{b}, \mathbf{a}, \mathbf{c}\}$, $\mathbf{S}^7 = \{\mathbf{b}, \mathbf{c}, \mathbf{a}\}$, $\mathbf{S}^8 = \{\mathbf{b}, \mathbf{c}, \mathbf{b}\}$. We train the RNN to learn the three-layer tree structure of the temporal orders of these sequences. We then design a new task of discriminating six temporal sequences formed by the above three primitive phonemes, which are "pcl-tcl-pcl", "pcl-tcl-pau", "pcl-pau-pcl", "pcl-pau-tcl", "tcl-pau" and "tcl-pcl", and their ordinal structure only cover part of the stored template in the RNN (shown in Fig.3C, denoted by red line). Following the same transfer learning procedure as described above, we train the model to learn new sequences, and observe that 1) the model can reuse part of the stored template to represent new sequences; 2) the model speeds up the learning process significantly compared to the case of no transfer learning (Fig.3D).

In addition to the aforementioned, we also conduct an additional experiment to demonstrate that our model can also combine and reuse primitive tree attractor structures to rapidly solve longer sequence task, shown in Fig. S8. Overall, our study shows that disentangled representation of ordinal structure in recurrent neural circuits can be efficiently combined and reused to facilitate transfer learning, if new temporal sequences have the same or partial ordinal structure.

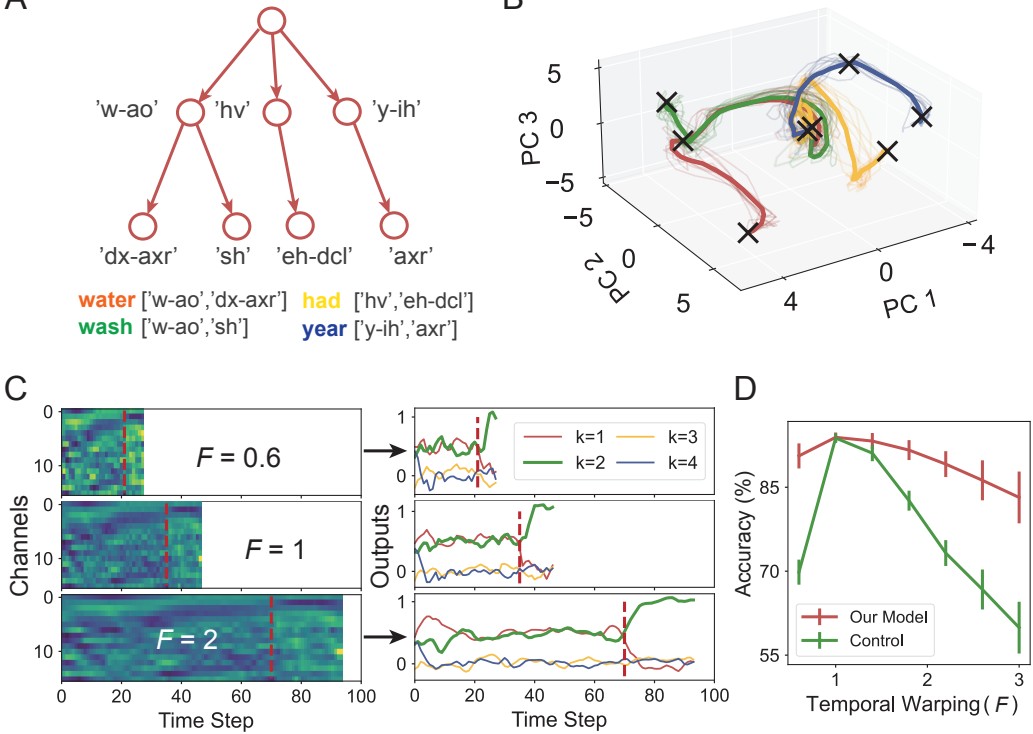

Figure 4: Robust key-word spotting with the ordinal structure. (A) The tree-structured orders of four words, 'water', 'wash', 'year', 'had'. (B) The visualization of the learned tree-structured neural representations of four temporal sequences in the reduced state space of the RNN. (C) Examples of the model discriminating the word 'water' with different warping effects ($F = 0.6, 1, 2$). Notably, the time for the model recognizing the word is also stretched or compressed accordingly. (D) With respect to temporal, our model achieves robust performances; whereas, a control model which does not follow the training protocol proposed in this work (and hence does not learn the tree-structured attractor dynamics) fails to do so. For the details of model parameters and performances, see Sec. D, F in SI.

## 3.2 Robust temporal sequence discrimination using ordinal structure

A hallmark of temporal information processing in the brain is that the neural system can recognize temporally stretched or compressed signals, known as temporal warping, as the same pattern [20]. It remains unclear how neural circuits in the brain achieves this goal. Here, we argue that storing ordinal structure as tree-structured attractor dynamics in recurrent neural circuits provides a feasible solution.

To test this hypothesis, we apply our model to a real-world application called key-word spotting. Specifically, we design a four spoken words discrimination task, where the four words, "water", "wash", "year", and "had", are chosen from the TIMIT dataset. For each word, 10 original temporal sequences are randomly sampled from the dataset, and each sequence is chunked into two primitive items by syllables or phonemes, for example, the word "wash" is chunked into "w-ao" and "sh". Overall, the temporal orders of these four words form a two-layer tree (Fig.4A). Following the same training protocol, we first augment training data by inserting varied intervals between neighboring primitive items and construct their target functions accordingly (for details, see Sec. A in SI). We then train the model to extract the ordinal structure (see Fig.4B). In testing, we present temporal sequences of four words without inserting separation intervals, rather we stretch or compress the original temporal sequences to reflect the temporal warping effect in practice (for details, see Sec. D in SI). We use a variable $F$ to denote the warping effect, with $F = 1$ representing no warping, $F > 1$ stretching, and $F < 1$ compressing. The results are shown in Fig.4B-D. We observe that: 1) the RNN extracts the ordinal structure of four temporal sequences; 2) the model realizes word discrimination

in a coarse-to-fine manner; 3) the mode achieves robust performance over a wide range of temporal warping.

## 4 Conclusion and Discussion

The brains is good at discovering and utilizing abstract representations, known as "schemas", from daily experiences. These schemas are suggested to be stored in the prefrontal cortex, and can be reinitialized and reused to afford efficient transfer learning to new tasks [21, 22]. However, there is still lack of deep understanding of how these schemas are represented and reused in neural circuits in the brain. In this work, we study the representation and reuse of a specific schema, the ordinal structure of temporal sequences, in neural circuits. We show that with an appropriate training protocol, a recurrent neural circuit can learn tree-structured attractor dynamics to represent the ordinal structure of temporal sequences. This abstract order template can then be bound with different contents, allowing for flexible and robust temporal sequence processing. We showcase this advantage by using a transfer learning and a key-word spotting tasks.

Our model may have some far-reaching implications to the representation of abstract schemas in the brain. Recently, researchers have conducted interesting studies on the format of neural schemas. For example, Guntupalli et al. proposed a graph model [23]. Similar to our work here, by learning new associations between new sensory features and schema representations, this graph model can utilize common abstract knowledge to facilitate transfer learning. However, their schema representations employ a high-order Markov dynamic model rather than the attractor dynamic model, which raises the question of how the brain biologically represents such schema representations. On the other hand, Goudar et al. trained a recurrent neural network model on various common sensor-motor mapping tasks and discovered a low-dimensional attractor dynamics which emerges and acts as an abstract structure to accelerate new decision-making tasks [24]. In contrast to Goudar's work, our research primarily focuses on abstract temporal order representation in the context of a real-world task involving key-word spotting. This task provides a more intricate and complicated setting for exploring abstract schema representations than what has been considered in previous studies. Overall, studying various attractor dynamics for different schema representations in the neural system would offer fascinating avenues for future research.

Although the focus of the present study is on unveiling the biological mechanism for representing the ordinal structure of temporal sequences, the computational principle we have gained can be applied to other brain-inspired computations. To test this idea, we replace the RNN in the model with two popular recurrent networks in machine learning, which are GRU and LSTM. Applying to the key-word spotting task, we observe that (Table.1): if the same training protocol proposed in this work is used, both GRU and LSTM achieve good performances; while if the conventional training protocol (i.e., no data augmentation and using the cross-entropy loss function) is used, they can still discriminate original temporal sequences but fail to be robust to temporal warping. These results suggest that our method (including the model structure and the training protocol) is general and has potential to be applied to brain-inspired computing.

Table 1: Model performances using different recurrent networks and different training protocols in a key-word spotting task. We replace the RNN in our model by GRU or LSTM. We train the model following either the same training protocol used in this work (marked by*) or the conventional protocol, i.e., no data augmentation and the cross-entropy loss function. Both models are trained on the dataset without warping ($F = 1$) and tested on a dataset with warping ($F = 0.6, 1.0, 1.4, 1.8, 2.2, 2.6, 3.0$). The classification accuracy of each model is obtained by averaging over $8$ trials with randomly split training and testing data, as well as network initialization. For each task, we have 10 samples per class in the train dataset and 300 samples per class in the test dataset. The word sequences are sampled from the TIMIT dataset. For more details, see Sec. D, F in SI.

| Model | 4 classes | | 8 classes | | 12 classes | |
|---|---|---|---|---|---|---|
| | no warping | warping | no warping | warping | no warping | warping |
| GRU | $94.0\% \pm 2.5$ | $88.9\% \pm 6.2$ | $95.1\% \pm 2.5$ | $87.9\% \pm 7.6$ | $94.3\% \pm 1.2$ | $84.4\% \pm 9.0$ |
| GRU* | $97.4\% \pm 1.5$ | $\mathbf{96.6}\% \pm 1.8$ | $97.2\% \pm 0.6$ | $\mathbf{95.6}\% \pm 1.8$ | $96.4\% \pm 0.9$ | $\mathbf{94.8}\% \pm 1.8$ |
| LSTM | $94.1\% \pm 2.2$ | $82.8\% \pm 13.7$ | $93.8\% \pm 1.6$ | $80.5\% \pm 12.3$ | $92.1\% \pm 0.7$ | $75.9\% \pm 14.8$ |
| LSTM* | $95.1\% \pm 1.8$ | $\mathbf{92.5}\% \pm 3.7$ | $94.0\% \pm 0.8$ | $\mathbf{90.1}\% \pm 3.5$ | $92.6\% \pm 2.1$ | $\mathbf{89.5}\% \pm 3.3$ |

In order to learn a tree-like structure in our model, we require precise ordinal details about continuous sequences. Several mechanisms can be employed in our brain to access this kind of information. Firstly, a large volume of experimental studies has shown that in the brain, continuous sequences are often chunked into discrete items to support high-level cognition [25]. For examples, speech sequences can be hierarchically chunked into words and syllables [1]; neurons in the hippocampus have been shown to detect event boundaries when watching continuous movie videos [26]. This chunking/segmentation process naturally outputs the ordinal structure of a temporal sequence. Additionally, computational models have been proposed for the brain performing sequence chunking, such as self-supervised learning [27] and oscillation [28]. Secondly, the development path of our brain also indicates the brain can naturally acquire sequence chunking. For example, in language acquisition, young children learn primitive phoneme categories around 2 months [29]; around 7 months, they learn sequences of either an ABB or ABA form [30], which can be represented as tree-like attractor structures in our model; subsequently, children learn to recognize spoken words around 12 months. Thus, in language acquisition, phoneme learning serves as a building block for word learning, which defines the ordinal structure of language sequences.

Incorporating a wide range of diverse temporal intervals presents another challenge for our model. How does the neural system handle this in a biologically plausible manner? Firstly, in our brain, motor and speech sequences are often generated with a large variability in speed, and hence they exhibit a large variability in separations between motor motifs and speech chunks. This large variability in separations enables the brain to learn the tree-like attractor structure, as demonstrated in our model. Secondly, in our network training, we do not really need very large intervals. For the clean synthetical data, we can actually train the tree-structured attractors using fixed interval values, shown in Fig. S10A. For the noisy spoken words, we do need an amount of variations to achieve good performances, but the range is only about 2 times of the item length. Overall, the requirement for a wide range of temporal intervals can be largely relaxed in practice.

Finally, there are several interesting directions worth exploring further. First, our current model is currently limited to small sequences. Figuring out how to effectively expand and scale our approach for more complex scenarios, like sequences involving large classes and longer sequences, will be a crucial focus for our research. Second, in the context of transfer learning, we employ a stochastic gradient descent method to learn the binding between sensory features and abstract representations. However, this technique is not biologically plausible and efficient. Experimental studies have shown the existence of a fast time-scale Hebbian rule in the brain, such as in the hippocampus [31], which is suggested to mediate rapid binding of distinct neural populations. By incorporating this fast Hebbian rule, our model could learn new associations between contents and the ordinal structure in an one-shot manner. Third, beyond ordinal structure and contents, the timing structure in a temporal sequence is also crucial for some tasks. Recent experiments have shown that the human brain could integrate abstract order template, timing structure, and contents to generate diverse finger action sequences [6]. Thus, integrating the timing structure into our model could potentially enhance its power for processing more complicated temporal sequence processing tasks. Last, the ordinal structure also serves as the basis for some more high-level abstract structures, such as algebraic patterns in motor sequence representation and recursive syntactic structures in language representation [1, 32]. How these abstract representations are concretely represented in neural circuits is largely unknown. Our model may serve as a starting point for exploring these mysteries.

## Acknowledgments and Disclosure of Funding

This work was supported by the National Natural Science Foundation of China (N0: T2122016, Y.Y.Mi), National Science and Technology Innovation 2030 Major Program (No. 2021ZD0203700 / 2021ZD0203705, Y.Y. Mi, No. 2021ZD0200204, Si Wu).

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

# Appendices

## A  The Training Protocol

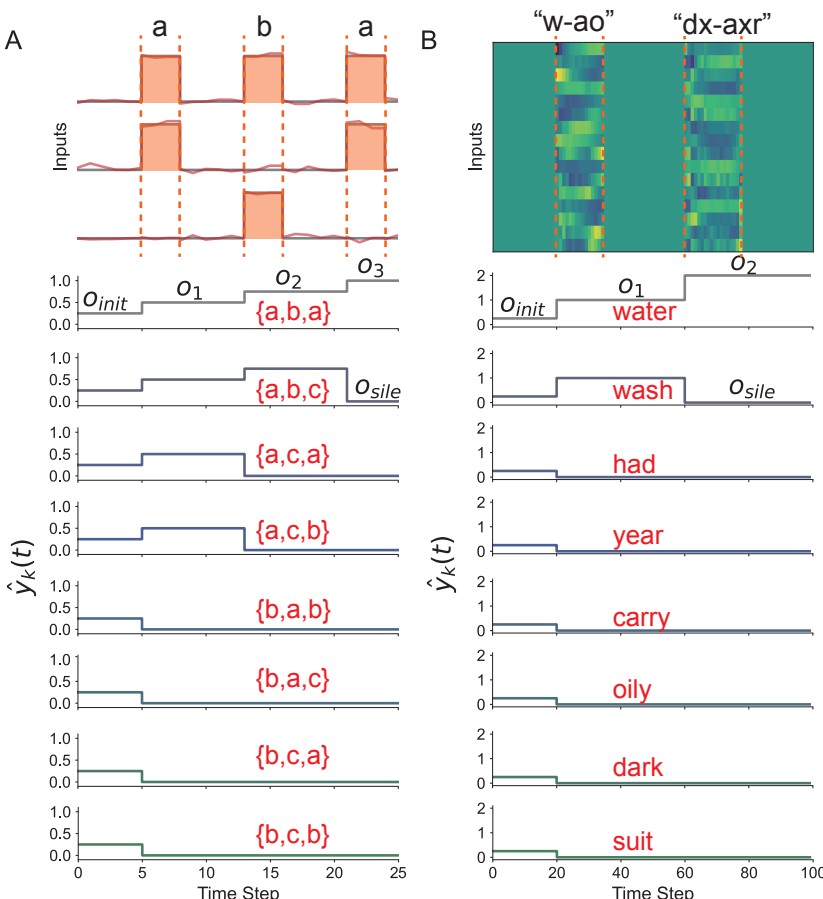

Figure S5: Illustration of the augmentation of temporal sequences and the setting of ramping target functions. (A) An example temporal sequence, $a - b - a$, in the synthetic task with $K = 8$, which is used to train the model shown in Fig.3C of maintext. The upper panel is augmented input stimuli by inserting duration and adding Gaussian noise, the lower panel is ramping target functions of readout neurons. (B) An example sequence, "water", in the key-word spotting task with $K = 8$ (used in Tab.1 of maintext ), which includes the follwing words: "water, wash, had, year, carry, oily, dark, suit". The upper panel is augmented MFCC feature vector of "water", the lower is the corresponding target functions of read-out neurons.

To train the model, we first augment training data and construct their target functions (see illustration in Fig.2A of maintext), after that we apply backpropagation-through-time (BPTT) algorithm to optimize the model parameters $\{\mathbf{W}^{in}, \mathbf{W}^{rec}, \mathbf{W}^{out}, \mathbf{I}^b\}$. Denote $\mathbf{y}(t)$ the actual model output sequence and $\hat{\mathbf{y}}(t)$ the corresponding target function. The loss function is given by

$$L = \frac{1}{T_b B} \sum_{b=1}^{B} \sum_{k=1}^{K} \sum_{t=1}^{T_b} \left[ y_k^b(t) - \hat{y}_k^b(t) \right]^2, \tag{3}$$

where $b = 1, \ldots, B$ denote the index of temporal sequences in the training data, $T_b$ the length of the $b$th temporal sequence, and $k = 1, \ldots, K$ the index of the output neuron. We adopt the ADAM optimizer, with the learning rate varying in the range of $lr = [1e^{-4}, 1e^{-2}]$, and the decay rates of the first and second moments are 0.9 and 0.999.

Inspired by the ramping activity of decision making neurons observed in the experiment [1–3], we set the target function of a read-out neuron increases with the unfold of the items in the temporal

sequence this neuron encodes. Consider the temporal sequence of the first class, denoted as $\mathbf{S}^1$, is presented to the network. The target functions of read-out neurons are constructed as below.

(1) The task of discriminating four synthetic temporal sequences ($K = 4$), Fig.2 in the main text.

In this task, $\mathbf{S}^1$ consists of two consecutive items $\{S_1^1, S_2^1\}$ with the duration $\Delta T$, as shown in Fig.2A-B in the main text. Before the temporal sequence is presented, all read-out neurons are at the spontaneous state, whose target functions are set to be $\hat{y}_k = O_{init}$, for $k = 1, ..., K$. When the first item $S_1^1$ appears, both the target functions of neuron 1 and 2 increase to $\hat{y}_{1,2} = O_1$, and the target functions of other neurons decay to the silent state, $\hat{y}_{3,4} = O_{sile}$. This lasts for the duration $\Delta T$, until the second item appears. When $S_2^1$ appears, the target function of neuron 1 further increases to $\hat{y}_1 = O_2$, while the target function of neuron 2 decreases to $O_{sile}$, and the target functions of other neurons keep the same, i.e., $\hat{y}_{2,3,4} = O_{sile}$. In this task, the parameters used are $O_{init} = 0.25$, $O_1 = 0.5, O_2 = 1, O_{sile} = 0$.

For other tasks of discriminating four temporal sequences, the target functions are designed similarly.

(2) The task of discriminating eight synthetic temporal sequences ($K = 8$), Fig.3C in the main text

In this task, $\mathbf{S}^1$ consists of three consecutive items $\{S_1^1, S_2^1, S_3^1\}$ with the duration $\Delta T$, as shown in Fig.3C in the main text. The design of target functions is illustrated in Fig.S1A, similar to the task with $K = 4$. Before the sequence is presented, all target functions of all read-out neurons are set to be $\hat{y}_k = O_{init}$ for $k = 1, ..., 8$. When the first item $S_1^1$ appears, the target functions of neurons $1 - 4$ increase to be $\hat{y}_k = O_1$, for $k = 1, 2, 3, 4$, since the four neurons share the same first item; while the target functions of other neurons decay to the silent state, i.e., $\hat{y}_k = O_{sile}$ for $k = 5, 6, 7, 8$. When $S_2^1$ appears, the target functions of neurons $1 - 2$ keep increasing to $\hat{y}_k = O_2$ for $k = 1, 2$; while the target functions of neurons $3 - 4$ decay to silent, and the target functions of other neurons keep silent, i.e., $\hat{y}_k = O_{sile}$, for $k = 3, ..., 8$. When $S_3^1$ appears, only the target function of neuron 1 keep increasing to $\hat{y}_1 = O_3$, while all other neurons are in the silent state, i.e., $\hat{y}_k = O_{sile}$, for $k = 2, ..., 8$. The parameters used are $O_{init} = 0.25, O_1 = 0.5, O_2 = 0.75, O_3 = 1$, and $O_{sile} = 0$.

For other tasks of discriminating eight temporal sequences, the target functions are designed similarly.

(3) The key-word spotting task of discriminating eight or twelve temporal sequences ($K = 8, 12$), Table 1 in the main text.

In this task, $\mathbf{S}^1$ denotes the temporal sequence of the key word 'water', which is chunked into two different items by syllables, i.e., 'w-ao' and 'dx-axr'. The training data is augmented according to the learning protocol described in the main text, as illustrated in Fig.S1B (upper panel). Before the sequence is presented, all read-out neurons are at the spontaneous state, i.e., $\hat{y}_k = O_{init}$, for $k = 1, ..., K$, with $K = 8, 12$. When the first item ('w-ao') appears, the target functions of neurons $1 - 2$ increase to $O_1$, while the target functions of other neurons decay to $O_{sile}$. When the second item appears, only the target function of neuron 1 keep increasing to $O_2$, while the target functions of other neurons are, $\hat{y}_k = O_{sile}$ for $k = 2, ..., K$. The design of target functions for $\mathbf{S}^1$ in 8-classes task is illustrated in Fig.S1B (lower panel). The parameters used are $O_{init} = 0.25, O_1 = 1, O_2 = 2$, $O_{sile} = 0$.

## B   The Visualization Method

### B.1   Principle Component Analysis (PCA) of neural activities in the RNN (Sec.2.4.1 in the Main text)

We apply PCA to find out the low dimensional state space which captures the major variance of neural activities in the RNN. Denote $\mathbf{r}^k(t)$ the activity vector of the RNN in response to the $k$th temporal sequence at moment $t$, for $k = 1, ..., K$, whose element is given by $r_i^k(t) = r_i(t)$, for $i = 1, ..., N$, with $N$ the number of neurons in the RNN. Combining the activity of the RNN at all moments, we obtain a martix $\mathbf{R}^k = \{\mathbf{r}^k(t)\}$, whose dimension is $N \times T$, with $T$ the length of $k$th temporal sequence. Define the covariance matrix $\mathbf{X} = \sum_{i=1}^{N_k} \sum_{k=1}^{K} (\mathbf{R}^k - < \mathbf{R} >)(\mathbf{R}^k - < \mathbf{R} >)^T$, where $< \mathbf{R} >$ is obtained by averaging activity vectors over time and the total testing trials, $K \times N_k$, with $N_k$ the number of trails in each class. We apply PCA to the matrix $\mathbf{X}$ and obtain a set of eigenvectors with decreasing eigenvalues. The eigenvalue of each PC reflects the variability of RNN response over time and trials along the corresponding eigenvector. As shown in Fig.2D in the main text, the first

3 PCs account for nearly $90\%$ of the variance of RNN activities. On the basis of this analysis, we project the RNN activities onto the first 3 PCs to visualize the evolution of the RNN state over time for different temporal sequences, as shown in Fig.2E in the main text.

## B.2 Stability analysis of tree-structured attractors (Sec.2.4.2 in the main text)

To uncover the structure of the state space of the learned RNN, we firstly find fixed points of the RNN dynamics, and then analyze their stability.

### (1) Finding fixed points in the RNN dynamics

Identifying fixed points or slow points in the dynamics of the RNN helps us to understand the dynamical properties of our model. Specifically, a fixed point is a state that remains unchanged over time to a constant input, while a slow point is a state that changes very slow over time to a constant input. The dynamics of the RNN is given by $\tau d\mathbf{x}(t)/dt = -\mathbf{x}(t) + \mathbf{W}^{rec}\mathbf{r}(t) + \mathbf{W}^{in}\mathbf{I}(t) + \mathbf{I}^b$, which can be approximately solved using the first order Euler method with the simulation time step $\Delta t = 1$, and the neural activity $\mathbf{r}(t)$ at time $t + 1$ can be approximately written as $\mathbf{r}(t + 1) = F(\mathbf{r}(t), \mathbf{I}(t+1)) = \tanh\left((1 - \Delta t/\tau) * \arctan\left(\mathbf{r}(t)\right) + \Delta t/\tau(\mathbf{W}^{rec}\mathbf{r}(t) + \mathbf{W}^{in}\mathbf{I}(t) + \mathbf{I}^b)\right)$, where $F(\cdot)$ is the state update function. Denote $\mathbf{r}^*$ a fixed point in the state space of the RNN, which satisfies the equation, $\mathbf{r}^* = F(\mathbf{r}^*, \mathbf{I})$ for a particular input $\mathbf{I}$. To get a better understanding of the network dynamics, we are also interested in slow points, which approximately satisfy $\mathbf{r}^* \approx F(\mathbf{r}^*, \mathbf{I})$.

To identify these fixed or slow points, we follow the approach developed by Sussillo and Barak [4] by solving an optimization problem. We focus only on the fixed and slow points when the network receives no external input (i.e., $\mathbf{I} = \mathbf{0}$). We define $q$ as an variable describing the speed at which the network state evolves over time, which is given by,

$$q = \frac{1}{2}||F(\mathbf{r}, \mathbf{I} = \mathbf{0}) - \mathbf{r}||^2. \tag{4}$$

The smaller $q$ is, the slower the network dynamics evolves. To optimize $q$ over the RNN state $\mathbf{r}$, we use BPTT and initialize from many conditions sampled from the distribution of network states during training. The neural states satisfying $q < 10^{-7}$ are considered to be fixed points. For the synthetic task as shown in Fig. 2 in the main text, we identify 7 fixed points in the RNN dynamics.

### (2) Stability analysis of fixed points

For each fixed point $\mathbf{r}^*$, we analyze its stability using the perturbation method [5]. Specifically, we apply perturbations to a fixed point $\mathbf{r}^*$, denoted as $\Delta\mathbf{r}(t)$ and $\Delta\mathbf{I}(t)$, respectively. The network state at $t + 1$ can be updated with the first-order Taylor expansion, which is given by,

$$\mathbf{r}(t + 1) = F(\mathbf{r}^* + \Delta\mathbf{r}(t), \mathbf{I}^* + \Delta\mathbf{I}(t)) \approx F(\mathbf{r}^*, \mathbf{I}^*) + \mathbf{J}^{rec}\Delta\mathbf{r}(t) + \mathbf{J}^{inp}\Delta\mathbf{I}(t), \tag{5}$$

where $\{\mathbf{J}^{rec}, \mathbf{J}^{inp}\}$ are Jacobian matrices given by $J_{ij}^{rec}(\mathbf{r}^*, \mathbf{I}^*) = \frac{\partial F(\mathbf{r}, \mathbf{I})_i}{\partial r_j}|_{(\mathbf{r}^*, \mathbf{I}^*)}$ and $J_{ij}^{inp}(\mathbf{r}^*, \mathbf{I}^*) = \frac{\partial F(\mathbf{r}, \mathbf{I})_i}{\partial I_j}|_{(\mathbf{r}^*, \mathbf{I}^*)}$.

The eigendecomposition of the recurrent Jocabian matrix $\mathbf{J}^{rec}$ can be expressed as $\mathbf{J}^{rec} = \mathbf{R}\mathbf{\Lambda}\mathbf{L}$, where $\mathbf{R}$ and $\mathbf{L}$ are matrices that contain the right and left eigenvectors of $\mathbf{J}^{rec}$, respectively, and $\mathbf{\Lambda}$ is a diagonal matrix containing complex-valued eigenvalues. The value of the real part of an eigenvalue determines the stability of the corresponding eigenmode, with those smaller than 1 representing stable modes and those larger than 1 representing unstable modes. As shown in Fig.S6, the maximums of real parts of all eigenvalues of each fixed point are all smaller than 1, indicating that all these fixed points are stable attractors of the RNN.

## B.3 Analyzing the transition dynamics of attractor states (Sec.2.4.3 in the main text)

We take the transitions from stable points $S_1^{1,2}$ to $S_1^2$ and $S_2^2$ as examples to illustrate the transition dynamics of the RNN. We combine the network activity trajectories in response to the $k$th class temporal sequence, for $k = 1, ..., K$, denoted as $\hat{\mathbf{R}}^k = \{\mathbf{r}^k(t_1 : t_2)\}$, to construct the matrix $\hat{\mathbf{X}}$ as in Sec.B.1 of SI, where $t_1$ and $t_2$ denote the moments of the appearance of item $S_1^{1,2}$ and the ending of the sequence, respectively. Define the covariance matrix $\hat{\mathbf{X}} = \sum_{i=1}^{N_k}\sum_{k=1}^{K}(\hat{\mathbf{R}}^k - <\hat{\mathbf{R}}>)(\hat{\mathbf{R}}^k - <\hat{\mathbf{R}}>)^T$, where $<\hat{\mathbf{R}}>$ is the average of network activities over time from $t_1$ to $t_2$ and over $K \times N_k$

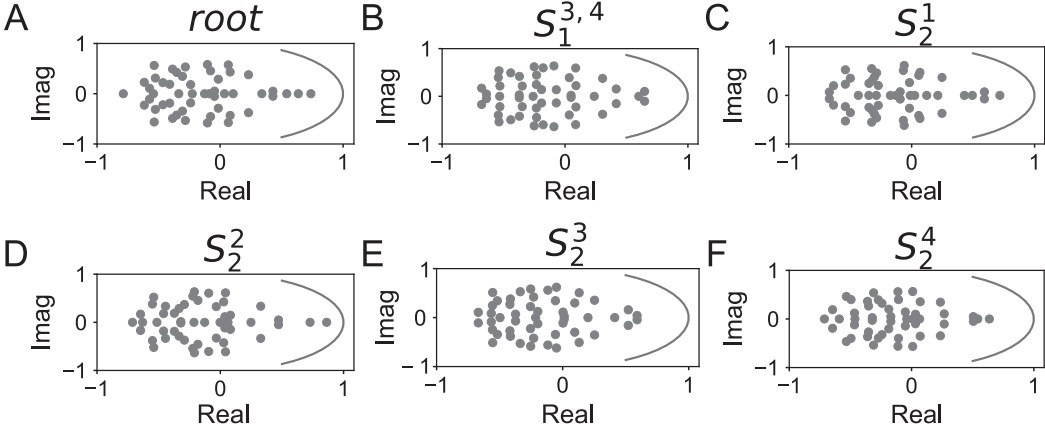

Figure S6: The distributions of complex eigenvalues for six attractor states of the RNN used in the synthetic task. The result for the other fixed point is shown in Fig.2D.

trails, with $N_k$ the number of trail in each class. We perform PCA on the matrix $\hat{\mathbf{X}}$ and obtain a set of eigenvectors with decreasing eigenvalues. The first two eigenvectors (denoted as $PC_1$ and $PC_2$) account for the major variance of neural responses.

We further analyze the right eigenvectors of the Jocabian matrix $\mathbf{J}^{rec}$ around the stable fixed point $S_1^{1,2}$. The larger the eigenvalue is, the more unstable the network activity state is along the eigenvector under noise perturbation. We project the right eigenvectors onto the hyperplane spanned by $PC_1$ and $PC_2$, with their lengths determined by the corresponding eigenvalues. We observe that the two most unstable directions among all eigenvectors point to nodes $S_2^1$ and $S_2^2$, as shown in Fig.2E-F. This indicates that it is much easier for the network state transferring from node $S_1^{1,2}$ to nodes $S_2^1$ and $S_2^2$ than to other states.

## C   Transfer Learning Tasks

### C.1   The construction of phoneme Sequences (Sec.3.1 in the main text)

The three primitive items, 'pcl', 'tcl' and 'pau' used in Fig.3 in the main text, are randomly selected from the TIMIT dataset [6]. We transform the raw waves of three phoneme items into Mel-frequency cepstral coefficients (MFCC) with a window size of 25ms and stride of 10ms, roughly mimicking the information processing in the cochlea in the brain. The obtained MFCC feature vectors have 16 channels, and the MFCC vectors are further normalized by subtracting their mean and dividing by their standard deviation, along the time dimension, which are used as inputs to the RNN. Using the four-class classification task as an example, as shown in Fig.2B in the main text, we construct four temporal sequences, which are "pcl-tcl", "pcl-pau", "tcl-pcl", and "tcl-pau". We augment these sequence by inserting varied intervals between primitive items and adding Gaussian noises. We then train the model using ramping target functions. In the task, the total number of augmented data is 10000, which are randomly divided into 8000 training and 2000 testing examples.

### C.2   The performance measurement (Fig.3A in the main text)

In transfer learning, the model learns to allocate phoneme contents to the pre-trained tree-like attractor template. To evaluate the effect of reusing the ordinal structure, we calculate the discrepancy between the learned attractor structure and the template, measuring by the Euclidean distance between them. Each item and its followed duration can be seen as a temporal chunk. If the model learns to reuse the pre-trained attractor dynamics, the last state of each temporal chunk should exactly stay at the corresponding attractor state. To test this, as an example, for the sequence $\mathbf{S}^1 = \{S_1^1, S_2^1\}$ in the four-class discrimination task, the learned state vector of $S_1^1$ item is obtained by averaging the state vectors over the last five time steps before the item $S_2^1$ is presented. Similarly, the learned state vector of $S_2^1$ item is obtained by averaging the state vectors over the last five time steps before the end of the sequence. The learned state vectors of other sequence items are computed similarly. After obtaining

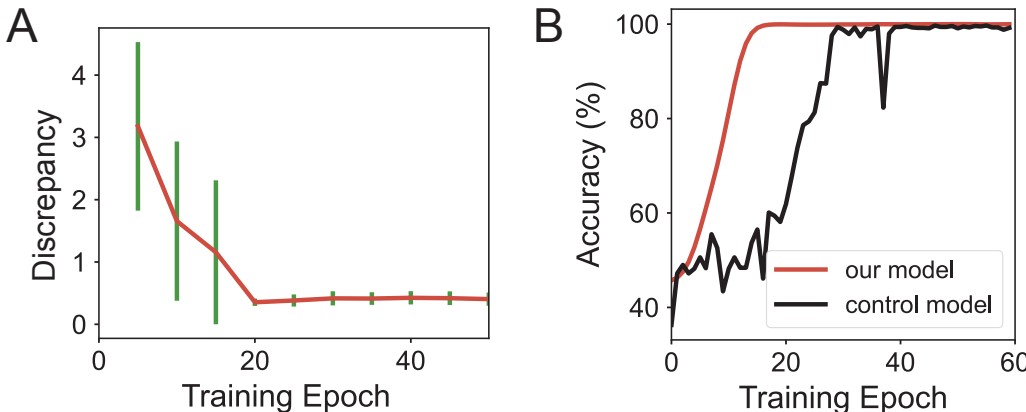

Figure S7: (A) The discrepancy between the learned dynamics and the template dynamics in the task of partially reusing the ordinal template. (B) In the transfer learning scenario, our model exhibits significantly accelerated learning compared to the control model using the cross-entropy loss.

these learned state vectors, we calculate the Euclidean distance between them and the corresponding tree-structured attractor state vectors in the template. The final discrepancy is computed by averaging the results of seven attractor states. As shown in Fig.3A and Fig.SS7, as the training progresses, the discrepancy between the learned attractor states and the template attractor states decreases gradually to zero, indicating that the model learns to allocate the input contents to the ordinal template stored in the RNN.

In transfer learning setting, We also compare our model with tree attractor structure to a pretrained recurrent neural network without using tree structrue. As shown in Fig. S7B, we train a control model having the same network structure as our model, except that the cross-entropy loss, rather than the ramping target function, is used. By this, the control model learns the sequence discrimination task but no longer acquires the tree-like attractors. We then apply the control model to the same transfer learning task (following the same protocol of freezing the connections in the recurrent and read-out layers). The results demonstrate that our model exhibits significantly accelerated learning process compared to the control model. This experiment demonstrates that the tree-structured attractor dynamics (the schema) is indeed indispensable for transfer learning. Together with our other analyses in Fig.3A and Fig. S7A), which verify that the attractor template is indeed reused after transfer learning, we come to the conclusion that the tree-structured attractor dynamics plays a critical role in transfer learning.

As shown in Fig. S8, we show that our model can learn to process sequences of depth 3 by combining two primitive templates of depth 2. In this experiment, for simplicity, we consider two independent recurrenct networks and there are no recurrent connections between them. Firstly, two RNNs learn and form same tree attractor structures of depth 2 through training on a four-class synthetical task. Then, following the transfer learning procedure as described in Sec.3.1, we freeze the recurrent connections in two RNNs and only optimize the feedforwad and readout connections, as shown in Fig. S8A. New eight-class phoneme task are constructed, written as "pcl-tcl-pcl", "pcl-tcl-pau", "pcl-pau-pcl", "pcl-pau-tcl", "tcl-pcl-tcl", "tcl-pcl-pau", "tcl-pau-tcl" and "tcl-pau-pcl". Our findings are as follows: 1) the pretrained attractor structures significantly accellerate the learning speed in a new sequence task, as depcited in Fig. S8C; 2) the network indeed combines and reuses two primitive tree attractor dynamics to facilitate the learning speed, as shown in Fig. S8D. These results demonstrate that our model has the capability to adapt to processing temporal sequences of varying lengths.

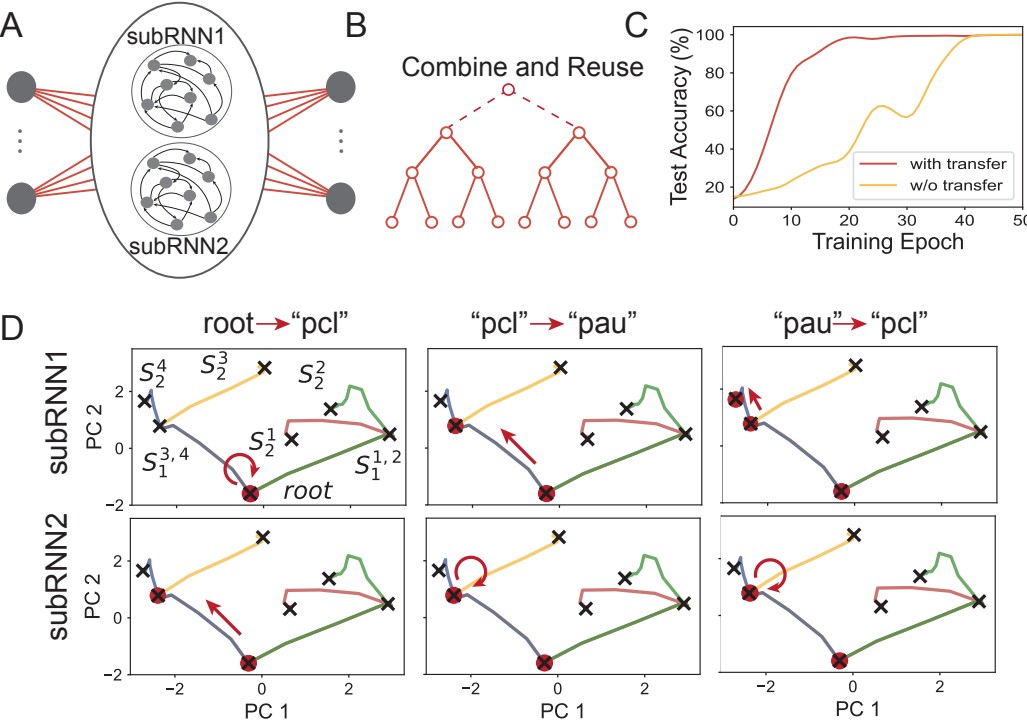

Figure S8: An example demonstrating primitive attractor template can be combined and reused to represent longer sequence. A. Network structure: two parts of a RNN (subRNN1 and subRNN2) encoding two depth- 2 tree-structured attractors. B. Schematic diagram: illustrating the combination and reuse of two tree- structured primitive templates. C. Significant acceleration in learning compared to the case without transfer learning. D. Attractor transition dynamics in the two subRNNs when a phoneme sequence "pcl"-"pau"-"pcl" is presented. The transition dynamics of subRNN1(the upper pannel) and subRNN2(the lower panel) are shown. In root-"pcl", when the network receives the 1st sequence item "pcl", subRNN1's state remain at the root node (red circles), while that of subRNN2 transits from the root node to $S_1^{3,4}$(shown in red arrow). In 1 "pcl"-"pau", when given the 2nd item "pau," subRNN1's state moves from the root node to $S_1^{3,4}$ while that 1 of subRNN2 stay at state $S_1^{3,4}$. In "pau"-"pcl", when given the 3rd item "pcl," subRNN1's state further 1 transits from $S_1^{3,4}$ to $S_2^4$. Thus, in transfer learning, the network combines two tree-structured attractor 12 templates to represent a longer sequence.

# D  Key-word spotting tasks

## D.1  Dataset Construction for Key-Word Spotting Task (Sec.3.2 in the maintext)

To construct the training and testing datasets for the key-word spotting task in Sec.3.2 in the main text, we randomly select twelve key words from the TIMIT dataset based on the word labeling. The TIMIT dataset comprises 6300 English sentences read by 630 speakers. The twelve key words are illustrated in Table.S1, which can be chunked into two different primitive items according to syllables. For example, 'wash' is chunked into 'w-ao' and 'sh'.

In the four-class key-word spotting task, we use the four words, "water, wash, had, year". In the eight-class task, we add another four words, "carry, oily, dark, suit". In the twelve-class task, we use all the selected words. In each task, the total number of temporal sequences for each word is nearly 310, which are randomly divided into 10 training and 300 testing examples. The raw wave of each temporal sequence is transformed into MFCC features using a time window size of 25ms and stride of 10ms. The obtrained MFCC vectors of each word is further normalized by subtracting their means and divided by their standard deviations along the time dimension.

In the training data, following our training protocol described in the main text, we chunk the MFCC vectors of a sequence into two primitive items according to the syllables of the key word, and then augment the data by inserting varied intervals between primitive items and adding Gaussian white noises. We then train the augmented temporal sequences of key words using ramping target functions, more details see Sec. A in SI. In the testing data, we present the MFCC vectors of each sequence without augmenting, rather we stretch or compress the MFCC sequences. The warping effect is specified by a variable $F$, with $F = 1$ denoting no warping, $F > 1$ stretching and $F < 1$ compressing.

Table S2: 12 words and their chunks, each chunk consists of phonemes or syllables.

| Words | Chunks |
|-------|--------|
| water | ['w-ao','dx-axr'] |
| wash | ['w-ao','sh'] |
| had | ['hv','eh-dcl'] |
| year | ['y-ih','axr'] |
| carry | ['kcl-k', 'eh-r-iy'] |
| oily | ['oy','l-iy'] |
| dark | ['dcl-d','aa-r-kcl-k'] |
| suit | ['s','ux-tcl'] |
| greasy | ['gcl','g-r-iy-s-iy'] |
| rag | ['r','ae-gcl-g'] |
| ask | ['ae','s-kcl'] |
| like | ['l','ay-kcl'] |

During training, both our model and the control model are trained on data with $F = 1$. During testing, we apply a warping factor $F$ to compress or stretch the original MFCC feature vectors using linear interpolation while retaining their spatial structure (particularly, we use $numpy.interp$ function to realize temporal warping, which works by finding the two nearest data points surrounding each value in the target data, constructing a straight line between these two points, and calculating the stretched or compressed target data along this line). After temporal warping, we scale the total strength of MFCC feature vectors by dividing a scaling factor $\sqrt{F}$, mimicking the divisive normalization process in the neural system [7]. Finally, the scaled versions of testing words are used to evaluate the model's classification robustness. Notably, scaling the strength of a MFCC feature vector slightly improves models' performance during testing, but in any case, our model outperforms other models significantly.

## D.2 Ablation study (Sec.4 in the main text)

In our training protocol, the introduction of varied temporal intervals between phonemes or syllables has the potential to generate spectral structures that closely resemble those present in the test examples. Consequently, the apparent efficacy of our model may be attributed to the exposure to training data that closely mirrors the test data, rather than leveraging tree attractor structures. To alleviate this concern and demonstrate the importance of the training protocol, we compare our model with several control models in the key-word spotting task. It demonstrates that the tree-structured attractor dynamics (the schema) contributes to the robustness of our model to warping sequences.

In our model, each training sequence is augmented to be 1.5 times of its original length by inserting varied intervals between phoneme or syllable chunks, and the model is trained using the ramping target function. In control model 2, training examples adopt the same augmentation protocol as in our model, but the model is trained using cross-entropy loss. In the control model 1, training examples are just warped sequences with the warping value chosen randomly in the range of $(1, 1.5)$. This makes the training and test data even more similar in the control model 1 than that in our model by data augmentation (in both models, the lengths of sequences are kept to be in the same range). We used the cross-entropy loss to train the control model 2. Thus, both control model 1 and 2 would not acquire proper tree-structured attractors.

We find that when the temporal warping value falls in the range of [1, 1.5], i.e., the range of training sequences, both models exhibit similar good test accuracies. However, when the warping value is outside of the training range, our model outperforms the control model significantly, as shown in Fig. S9.

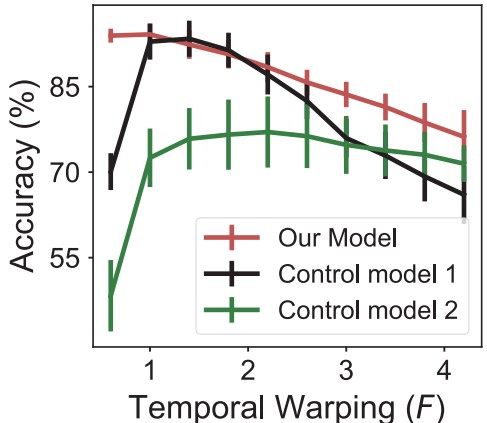

Figure S9: Performance comparison of our model with two control models. In our model, the training data is augmented by introducing varied intervals between "chunks" of a given word. Each sequence is augmented to be 1.5 times of its original length. Following the same augmentation protocol as in our model, Control model 2 (green line) is trained using the cross-entropy loss. In Control model 1(black line), the cross-entropy loss is used, and the training data are warping sequences with the warping value chosen randomly in the range of (1, 1.5). The test accuracy of each model is calculated by averaging over 8 trials at each warping value $F$.

This supports our statement that the tree-structured attractor dynamics (the schema) contributes to the robustness of our model to warping sequences. The underlying mechanism can be attributed to two factors: 1) attractors ensure the stable responses of our model to stretching/compressing inputs; 2) the tree-structured attractor dynamics further enhances averaging out noises in inputs over time, behaving like an evidence-accumulation process (see Fig.4C)

We also apply our training protocol to other recurrent networks used in machine learning, including Long Short-Term Memory (LSTM) [9] and Gated Recurrent Unit (GRU)[10]. Using the datasets in the key-words spotting task, both GRU and LSTM are trained by using either our learning protocol or the conventional learning method (which involves data augmentation and the cross-entropy loss function). The results are compared in Table.1 in the main text.

## E   Experiments with different training Settings and model architectures

We find that the tree-structured attractors also emerge when training using fixed interval values for the clean synthetical data, as shown in Fig. S10A. However, for the noisy spoken words, we do need an amount of variations to achieve good performances.We also examine our method on GRU network to test its university across model architectures. As shown in Fig. S10B, GRU network can also learn a tree-structured attractors successfully.

## F   Model Parameters used in this study

All models are trained using PyTorch 2.0 [8].

(1) Parameters in Fig.2 in the main text:

The parameters of the RNN are: $M = 3$, $N = 50$, $K = 4$, $\tau = 2$. The parameters for data augmentation are: $\Delta t = 3$, $\Delta T_{max} = 30$, the noise strength inserted between neighboring items is $\sigma = 0.01$. The recurrent connection weights $\mathbf{W}^{rec}$ are initialized by sampling from an independent Gaussian distribution with zero mean and SD equal to $g/\sqrt{N}$, with $g$ representing the 'gain' of the network, $g = 0.2$. For the design of target functions, see Sec. A. The batch size is $B = 16$, and the total number of training epochs is 100.

(2) Parameters in Fig.3 in the main text:

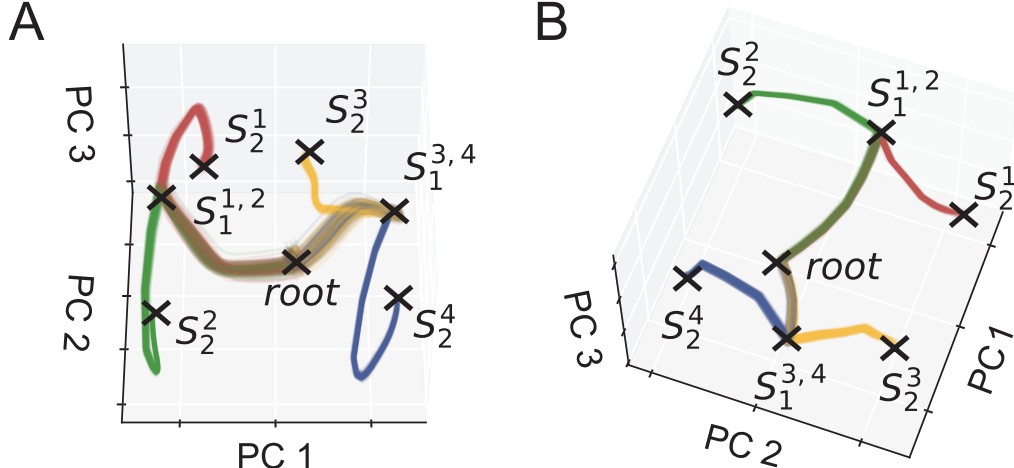

Figure S10: (A) In a 4-class synthetic sequence task, our model successfully learns a tree-like attractor structure from sequence inputs with a fixed interval of 10 time steps. (B) A GRU model trained using our method also learns a tree-like attractor structure on the 4-class synthetic task.

The parameters of the RNN are: $M = 16$, $N = 50$, $K = 4$ in Fig.4A-B and $K = 8$ in Fig.4C-D, $\tau = 2$. The parameters for data augmentation are: $\Delta T_{max} = 30$, the noise strength inserted between neighboring items is $\sigma = 0.01$. For details of the construction of phoneme temporal sequences, see Sec. C in SI. $\mathbf{W}^{rec}$ is initialized by sampling from an independent Gaussian distribution with zero mean and SD equal to $g/\sqrt{N}$, $g = 0.2$. For the design of target functions, see Sec. A in SI. The batch size is $B = 16$, and the total number of training epochs is 100.

(3) Parameters in Fig.4 in the main text:

The parameters of the RNN are: $M = 16$, $N = 100$, $K = 4$, $\tau = 2$. The parameters for data augmentation are: $\Delta T_{max} = 60$, the noise strength inserted between neighboring items is $\sigma = 0.01$. The recurrent connection weights $\mathbf{W}^{rec}$ are initialized by sampling from an independent Gaussian distribution with zero mean and SD equal to $g/\sqrt{N}$, with $g$ representing the 'gain' of the network, $g = 0.2$. For the design of target functions, see Sec. A. The batch size is $B = 16$, and the total number of training epochs is 200.

(4) Parameters in Table 1 in the main text:

The parameters of LSTM, GRU and RNN are: $M = 16$, $N = 100$, $K = 4$. The time constant of neurons in the RNN is $\tau = 2$. The parameters for data augmentation are: $\Delta T_{max} = 60$, the noise strength inserted between neighboring items is $\sigma = 0.01$. The recurrent connection weights $\mathbf{W}^{rec}$ are initialized by sampling from an independent Gaussian distribution with zero mean and SD equal to $g/\sqrt{N}$, with $g$ representing the 'gain' of the network, $g = 0.2$. The connections of LSTM and GRU adopt Glorot are initialized by an uniform distribution. For the design of target functions, see Sec. A. The batch size is $B = 16$, and the total number of training epochs is 200.

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
