# OpenReview forum: "Learning and processing the ordinal information of temporal sequences in recurrent neural circuits"
_NeurIPS.cc/2023/Conference — NeurIPS 2023 poster_

### Official Review · Reviewer_CeTG · 2023-06-29

**Soundness:** 3 good
**Presentation:** 2 fair
**Contribution:** 2 fair
**Rating:** 6
**Confidence:** 4

**Summary:**

In this manuscript the authors use a custom training regime to force simple recurrent neural networks (RNNs) to learn the ordinal structure of sequential inputs. Specifically, they train the network to learn the order of sequences by presenting the elements of a sequence with variable durations and variable intervals between elements. They demonstrate the utility of these networks in both a toy transfer learning task and a more realistic "key-word spotting" task.

**Strengths:**

Strengths:

1. The manuscript presents the key ideas in a straightforward and lucid manner
2. The core idea – training RNNs to recognize the ordinal structure of its inputs through varying the duration of its constituent elements – is interesting with potential downstream applications in machine learning and computational neuroscience


**Weaknesses:**

Weaknesses:

Major:

1. The results in Figure 3 are difficult to interpret without a more systematic bank of control models. For instance, in the transfer learning task, as far as I can tell the authors compare transfer learning of their particular model to a model with no transfer learning. This does not tell us if the tree structure learned by their model is doing the heavy lifting. Specifically, have the authors compared their "tree-structure" model to an RNN that learns the task without using tree structure? Without this key control we cannot assess whether tree structure is what confers performance in this task, or if it is simply a matter of their model having been trained to solve *any* task prior to transfer learning.
1. I am a bit confused about what Figure 4 is showing. Here, the authors train their network using varied intervals between "chunks" of a given word. One issue that I can see with this training regime is that, in addition to enforcing tree structure in network dynamics, it is also presenting a "warped" version of the inputs. In other words, does inserting a delay between phonemes create spectral structure more similar to the test inputs than the phonemes *without* gaps? If true, this would imply that the network is simply seeing training data more similar to the test data, rather then leveraging tree structure. Further controls and analysis would alleviate this concern.

Minor:

1. Typos throughout, the manuscript needs a pass on the writing.

**Questions:**

1. Can the authors provide more detail about the control models in Figures 3 and 4? My concerns could be simple misunderstanding.

**Limitations:**

Limitations are adequately addressed.

---

> ### Author Rebuttal · Authors · 2023-08-09
>
> Thanks for the valuable comments, which are very helpful for us to improve the paper. Below are our detailed replies.
>
> Weaknesses
>
> **On the contribution of tree-structured attractors**
>
> Thank for raising this concern. To address this concern, we conducted an additional experiment (see Fig.R2-C in the uploaded reply PDF), in which we trained a control model having the same network architecture, except that the cross-entropy loss, rather than the ramping target function, was used. By this, the control model learned the sequence discrimination task, but no longer acquired the tree-structured attractors. We then applied the control model to the same transfer learning task (following the same protocol of freezing the connections in the recurrent and read-out layers). As shown in Fig.R2-C, our model exhibits significantly accelerated learning speed compared to the control model. This experiment demonstrates that the tree-structured attractor dynamics (the schema) is indispensable for fast transfer learning. Together with our other analyses in Fig.3A and Fig.S3 (which verify that the attractor template is indeed reused during transfer learning), we conclude that the tree-structured attractor dynamics is critical for fast transfer learning.
>
> **On the control of training data in warping performance**
>
> Thanks for raising this concern. To clarify this concern, we conducted an additional experiment, in which we built a control model which just uses warped sequences as the training data. This makes the training and test data even more similar than that in our model. We adopted the cross-entropy loss to train the control model, such that the recurrent network did not acquire proper tree-structured attractors. We found that when the temporal warping value falls in the range of [1, 1.5], i.e., the range of the lengths of training sequences, both models exhibit similar good test accuracies. However, when the warping value is outside of the training range, our model outperforms the control model significantly (see Fig.R2-D in the uploaded reply pdf). This control experiment excludes the possibility that the similarity between training and testing data leads to the good performance of our model, and it supports that the tree-structured attractor dynamics contributes to the robustness of our model to warped sequences. The underlying mechanism can be attributed to two factors: 1) attractors ensure the robust responses of our model to stretching/compressing inputs; 2) the tree-structured attractor dynamics further average out noises over the sequence, behaving like an evidence-accumulation process (see Fig.4C).
>
> Minor
>
> We will thoroughly improve the writing in the revised manuscript.
>
> Questions
>
> **On the control models in Figures 3 and 4**
>
> In Fig.3, the control model shares the same network architecture, data augmentation and target function as our model. The only difference is on the training protocol. In our model, the connections in the recurrent layer, which store the tree-structured attractors, are frozen during the transfer learning, and only feedforward connections are updated; whereas, in the control model, both recurrent and feedforward connections are re-trained (more details see lines 209-214 in the supplementary materials). This comparison allows us to demonstrate the effect of the schema on accelerating transfer learning.
>
> In Fig.4, the control model shares the same network architecture as our model. In the control model, no data argumentation is used, and the cross-entropy loss is adopted. By this setting, the control model can learn a given set of sequences, but does not form the tree-structured attractor dynamics. This comparison is to demonstrate that the tree-structured attractor dynamics enhances the robust of our model to warped sequences. However, as pointed out by the reviewer, since no data argumentation is used in the control model, it does not exclude the possibility that the improved performance of our model comes from data argumentation, as it tends to make the training and testing data more similar. To clarify this concern, we train another control model as described above (see Fig.R2-D in the reply PDF), which directly uses warped sequences as training examples, and hence it makes the training and test data even more similar than our model. Again, we observe that our model outperforms the control model to unseen warped sequences. This further strengthens our conclusion that the learned schema facilitates the robustness of our model to warped sequences.
>
> We will expand the descriptions about control models (including the new ones) in the revised manuscript to make them more clearly.
>
> We hope that we have addressed all the concerns of the reviewer and could convince the reviewer to raise the score.

---

> > ### Comment · Reviewer_CeTG · 2023-08-14
> >
> > The authors have adequately addressed all of my concerns, and I have raised my score accordingly.

---

> > > ### Author Response · Authors · 2023-08-14
> > >
> > > Thank you for your improved score and positive feedback on our paper. We greatly appreciate your comments as they help us to further refine our work !

---

### Official Review · Reviewer_dJ2Y · 2023-07-06

**Soundness:** 3 good
**Presentation:** 3 good
**Contribution:** 2 fair
**Rating:** 6
**Confidence:** 3

**Summary:**

This paper investigates how recurrent neural circuits learn to represent the abstract order structure of temporal sequences and how the disentanglement facilitates sequence processing. The main objectives were better understand the brain's mechanisms for representing temporal sequence ordinal information and contributing to the development of brain-inspired sequence processing algorithms.

The authors found that given suitable training, a recurrent neural circuit can learn tree-structured attractor dynamics to encode the tree-structured orders of temporal sequences. They show that reusing a temporal order template aids the learning of new sequences sharing the same or partial ordinal structure and that the tree-structured attractor dynamics improve the robustness of temporal sequence discrimination.



**Strengths:**

The authors demonstrated that in a supervised learning task, recurrent neural circuit can learn tree-structured attractor dynamics to encode the corresponding tree-structured orders of temporal sequences. They used a transfer learning task to show that once the network has learned the temporal structure, it can apply that knowledge to different temporal inputs - this was demonstrated by freezing the recurrent weights and training only weights in a feedforward layer that followed the recurrent layer. They also showed that data augmentation can lead to invariance to temporal rescaling. These results are consistent with several neuroscience studies.

**Weaknesses:**

The evaluation is relatively limited and uses only short sequences and supervised learning. If the brain indeed uses a similar mechanism, then it should successfully scale to much longer sequences.

If the model is capturing cognitive properties of human sequence memory, then it should also be consistent with results from behavioral studies, which are characterized by effects such as primacy, recency and contiguity.

The authors might consider reflecting on or comparing their approach with other work related to modeling sequences in the brain, for example:

Cui et al. 2016 Continuous online sequence learning with an unsupervised neural network model

Graves et al. 2014 Neural Turing machines

Voelker et al. 2019 Legendre memory units: Continuous-time representation in recurrent neural networks

Eliasmith et al. 2013 A large-scale model of the functioning brain

Howard et al. 2014 A unified mathematical framework for coding time, space, and sequences in the hippocampal region

Whittington et al. 2020 The Tolman-Eichenbaum machine: unifying space and relational memory through generalization in the hippocampal formation

Also, for recent work on biologically inspired neural networks robust to temporal rescaling see

Jacques et al. 2022 A deep convolutional neural network that is invariant to time rescaling



**Questions:**

When you used data augmentation and varied the duration between neighboring items, how many steps of BPTT did you have to do and have you encountered issues with vanishing/exploding gradients?

The authors reflected on the biological implausibility of backpropagation and proposed the possibility of using the fast Hebbian rule. It is not clear whether this could work in the context of backpropagation through time since it would require that the biological system stores a copy of the temporal input. Do the authors have a possible solution for this?

Do you think the observed properties could emerge if the training was done in a self-supervised way?

**Limitations:**

The authors addressed some of the limitations, especially regarding the backpropagation (but see my earlier comments on how I suggest expanding those).

---

> ### Author Rebuttal · Authors · 2023-08-09
>
> Thanks for the valuable comments, which are very helpful for us to improve the paper. Below are our detailed replies.
>
> Weaknesses
>
> **On the scalability of the model**
>
> Thanks for raising this important issue. As the first step of presenting the framework, we only evaluate the model with short sequences. The study of long sequences will be added in future work. Here, we would like to point out that our model has the capacity of processing long sequences, and it can achieve this in two biologically plausible ways.
>
> First, the brain can combine short primitive templates to form long ones. To demonstrate this idea, we conduct a preliminary experiment, in which the network learns to represent sequences of length 3 by dynamically combing two shorter templates of length 2 (see Fig.R1 in the uploaded reply pdf).
>
> Second, the brain can employ a hierarchical network to combine ordinal templates in each layer hierarchically. Notably, this way of combining primitive templates to encode long sequences was proposed by Chomsky for language processing, the so-called the minimalist program [1]. The program suggests that our human brain is capable of processing arbitrarily complex and lengthy language sentences through recursive dynamic binding of primitive linguistic units.
>
> **On the cognitive properties of human sequence memory**
>
> Thanks for the suggestion. The effects of primacy, recency, and contiguity are often reported in sequential working memory tasks, and a potential computational mechanism is the short-term plasticity (STP) of synapses [2]. Here, our focus is on studying the learning of tree-structured attractors (the schema), but we presume that if we also consider a sequential working memory task and include STP in feedforward connections, we should observe similar effects.
>
> Alternatively, we may compare our model to those schema-related experimental studies. For example, a recent study [3] found that rats could store the odor sequence structure as a schema by using a low-dimensional neural code in the orbitofrontal cortex, and this schema facilitated learning of new similar tasks. Our model also shows that the ordinal structure is encoded as tree-structured attractors in a low-dimensional space, and this attractor dynamics facilitates transfer learning. In future work, we will compare our model with the experimental data in details.
>
>
> **On the differences to other papers**
>
> Thanks for providing these references. We checked all of them and summarized their differences to our work as below:
>
> The works by Cui et al. (2016), Graves et al. (2019), and Eliasmith et al. (2013) addressed sequence modeling using different ways, such as HTM with an unsupervised Hebbian rule, external memory modules, and cognitive architectures. These models differ from our work on that they do not explicitly disentangle ordinal structures from contents. Whittington et al. (2020) proposed TEM to disentangle structure from contents and integrate them via conjunctive coding. The self-supervised method used by TEM may learn a tree-like attractor structure in an unsupervised manner. However, TEM does not tell how to store the learned sequence structures and reuse them for new tasks. The LMU model by Voelker et al. (2019) efficiently captures long temporal dependencies. Our method can be applied to train the LMU model for learning complex tree-like structures, and potentially further leverage the LMU's efficiency in capturing long dependencies. Jacques et al. (2022) proposed a new deep convolution network that utilizes logarithmically compressed temporal representations, but does not consider the extraction of disentangled ordinal structure.
>
> Overall, our model is fundamentally different from these works on that we explore how the brain learns the disentangled ordinal structure (schema) by employing tree-structured attractor dynamics, and how this schema facilitates transfer learning. Nevertheless, our model is not contradictory to these works, rather they can be integrated together to perform complex sequence tasks.
>
> Questions:
>
> 1. We typically performed 90 to 150 steps of backpropagation through time (BPTT). To mitigate the potential vanishing or exploding gradients, we employed gradient clipping and carefully initialized the weights. Thus, we did not encounter vanishing/exploding gradients in practice.
>
> 2. As the fast Hebbian rule is differentiable, it can be applied in the context of backpropagation through time (BPTT). Recent works in both neuroscience and machine learning society have explored this issue [4,5,6,7]. In our model, we can replace the static feedforward connections from inputs to the recurrent network with context-controlled fast Hebbian weights and train the model using meta learning methods [8]. In such a way, context information can modulate feedforward connections, enabling fast binding or rebinding between contents and ordinal templates. Thus, the integration of fast Hebbian rule and BPTT may facilitate the learning and utilization of the tree-structured attractor dynamics.
>
> 3. Yes, it is possible that the tree-like attractor structure emerges in the network through self-supervised learning. A possible solution is that: we first apply a self-supervised method [9] or a quantized representation method [10] to chunk raw input sequences into discrete items; we then apply a self-supervised learning method, such as the contrastive predictive loss [11], to train the recurrent network to get the tree-structured attractor dynamics.
>
> References:
>
> [1]	N. Chomsky, MIT Press, 1995
>
> [2]	Mi et al, Neuron, 2017
>
> [3]	Zhou et al, Nature, 2021
>
> [4]	Ba et al, Neurips, 2016
>
> [5]	Thangarasa et al, ICML, 2019
>
> [6]	Tyulmankov et al, Neuron, 2022
>
> [7]	Dekker et al, PNAS,2022
>
> [8]	Wang et al, Current Opinion in Behavioral Sciences, 2021
>
> [9]	Asabuki et al, Nature communications, 2020
>
> [10]	Van Den Oord et al, Neurips, 2017
>
> [11]	Oord et al, arXiv, 2018

---

> > ### Comment · Area_Chair_1YLr · 2023-08-17
> > **Reviewer response needed**
> >
> > Hello Reviewer,
> >
> > The authors have endeavoured to address your comments in their rebuttal. The rebuttal phase is a key part of the NeurIPS review process. I invite you to read and respond to the author's comments as soon as possible, latest tomorrow, to give everyone time to continue and conclude the discussion.
> >
> > Thank you for helping make NeurIPS a great conference for our community.

---

> > ### Comment · Reviewer_dJ2Y · 2023-08-19
> >
> > Thank you for providing a detailed response. I don't have any additional questions.

---

### Official Review · Reviewer_tPSE · 2023-07-07

**Soundness:** 3 good
**Presentation:** 4 excellent
**Contribution:** 2 fair
**Rating:** 4
**Confidence:** 4

**Summary:**

This paper describes a method for training RNNs that is used to extract ordinal sequences.  There are two variations on the training that make this possible.  First, the network is trained on sequences with a wide range of temporal delays, so that only ordinal position is relevant.  Second, the training signal is the location of each sequence on a tree structure given a priori to describe the set of sequences.

The model is tractable and interpretable and two classes of findings are described.  First, networks that have learned a tree structure for a particular problem can generalize rapidly to new problems with the same structure by freezing the recurrent weights and relearning the output weights.  Second, the model is tested on time-warped versions of a set of spoken words and it generalizes better than a control model.

**Strengths:**

The exposition is extremely clear.



**Weaknesses:**

The requirement that the training set provides information about the true ordinal structure seems very strict.

The requirement for training on a wide range of temporal intervals is a serious limitation as a model of the brain.

I find the colored lines in Fig 1A,C very difficult to distinguish.

**Questions:**

Under what circumstances would we expect the network has access to a training signal with information about the ordinal position in a tree structure?  What is the use case for this network?  How could one discover that structure without training it in?

A simple way to build in a model of ordinal position is to have an RNN where the rhs is modulated by a gating factor:
$
dx/dt = \alpha(t) [ \ldots ]
$
where $\alpha(t)$ can be learned.  If $\alpha(t) = 0$ between relevant triggering stimuli, one can say it's learned an ordinal code.  Is that possible in a GRU?

How does this model compare in its time warping performance to classic algorithms (e.g., Sekoe & Chiba, 1978)?  On a related note, a recent approach (Jacques, et al., 2022, ICML) shows effectively perfect generalization over a wide range of warping factors without data augmentation.  How does this model relate to that work?

---

> ### Author Rebuttal · Authors · 2023-08-09
>
> Thanks for the valuable comments, which are very helpful for us to improve the paper. Below are our detailed replies.
>
> Weaknesses
>
> **On the prior of the ordinal structure**
>
> Thank for raising this concern, but we would like to point out that this is not a problem for the brain, although it may be a concern for machine learning tasks.
>
> First, experimental studies have shown that in the brain, continuous sequences are chunked into discrete items to support high-level cognition [1]. For examples, speech sequences can be hierarchically chunked into words and syllables [2]; neurons in the hippocampus have been shown to detect event boundaries when watching movie videos [3]. This chunking process naturally produces ordinal structures of temporal sequences. Computational models are also proposed for neural chunking, such as self-supervised learning [4] and oscillation [5].
>
> Second, the development path also indicates that the brain implements sequence chunking. For example, in language acquisition, young children learn primitive phonemes around 2 months [6], learn sequences of either ABB or ABA form around 7 months [7], and learn to recognize spoken words around 12 months. Here, phoneme learning serves as a building block for word learning, defining the ordinal structure of language sequences.
>
> Since our focus is on exploring how the brain learns the ordinal structure disentangled from contents, the so-called schema which is receiving increasing attention in theoretical neuroscience [8], we assume that the input sequences to our model have already been chunked in advance. While when applying our model to machine learning, it does need a method to first segment sequences based on the statistics of items.
>
> **On the wide range of temporal intervals**
>
> Thank for raising this concern, but we would like to argue that this is not a serious limitation to the brain.
>
> First, in the brain, motor and speech sequences are generated with large variability in speed, and they exhibit large variability in separations between motor motifs and speech chunks. This large variability enables the brain to learn the tree-like attractor structure.
>
> Second, in our network training, we actually do not need very large intervals. For the clean synthetical data, we can actually train the tree-structured attractors using fixed intervals (see Fig.R2-A in the reply pdf). For noisy spoken words, we do need an amount of variations in intervals to achieve good performance, but the range is only about 2 times of the item length. Overall, it does not need the range of temporal intervals to be very large.
>
> Questions
>
> **On the available of ordinal structure, the use of the model, and un-supervised learning**
>
> 1) As replied above, the brain can access the ordinal structure through chunking. Experimental data has shown that continuous sequences such as motor and speech sequences are hierarchically chunked to form discrete sequence representations.
>
> 2) Our network can be applied to model the learning of abstract ordinal structures of temporal sequences (schema), such as motor sequences, speech, and language. For example, a potential application is to model language appreciation (certainly many more works are needed), where primitive ordinal structures are stored to facilitate language understanding. Our model may also be applied to solve transfer learning in machine learning.
>
> 3) To discover sequence structures without supervised signals, we may mimic the chunking mechanism in the brain. Incorporating a generative objective in unsupervised learning may also aid in extracting sequence structures. The chunking mechanism may use a self-supervised loss [4] or quantized representation trick [9]. Generative objective loss may employ the contrastive predictive coding method [10] or some other self-supervised methods.
>
> **On GRU**
>
> Sorry, we do not get the meaning of the reviewer, “A simple way to build in a model of ordinal position is to have an RNN where the rhs is modulated by a gating factor…”.
> In term of GRU, it can indeed learn the tree-like attractor structure if our learning protocol is used. The performance of GRU after learning tree-like attractors is shown in Table 1. We can visualize its tree-like attractor structure of neural activities in GRU, see Fig.R2-B in the reply PDF.
>
> **On the warping performance**
>
> Thanks for pointing out these two references we have missed. We did not evaluate our model intensively on the warping performance. This is because that the goal of the current paper on exploring how the brain learns the disentangled ordinal structure (schema). Since the key of schema is on transferring learning, we have focused on investigating this issue. In future work, we will systematically evaluate the warping performance of our model and compare it with other methods.
>
> Considering the differences to the two works: 1) Sekoe & Chiba proposed a machine learning kind of method for representing the ordinal structure, while our method focuses on the neural representation of the ordinal structure; 2) the model of Jacques et al. was inspired by time cells in the brain, while our model was inspired by the disentangled representation of ordinal structure, manifested as tree-structured attractors. Jacques’ model may perform well for recognizing a given set of warping sequences, it likely has difficulty for transfer learning, since it does not extract the schema as done in our model.
>
> We hope that we have addressed all concerns of the reviewer and could convince the reviewer to raise the score.
>
> References:
>
> [1]	Dehaene et al, Trends in Cognitive Sciences, 2022
>
> [2]	Dehaene et al, Neuron, 2015
>
> [3]	Hahamy et al, Nature neuroscience,2023
>
> [4]	Asabuki et al, Nature communications, 2020
>
> [5]	Giraud et al, Nature neuroscience,2012
>
> [6]	Kuhl et al, Nature reviews neuroscience,2004
>
> [7]	Marcus et al, Science,1999
>
> [8]	Goudar et al, Nature Neuroscience,2023
>
> [9]	Van Den Oord et al, Neurips, 2017
>
> [10]	Oord et al, arXiv, 2018

---

> > ### Comment · Reviewer_tPSE · 2023-08-10
> >
> > I have read and considered the author's response.

---

### Official Review · Reviewer_7Qop · 2023-07-07

**Soundness:** 4 excellent
**Presentation:** 3 good
**Contribution:** 4 excellent
**Rating:** 5
**Confidence:** 4

**Summary:**

The canonical biological neural circuit model, described by equation (1) in this work, primarily relies on attractor dynamics to perform cognitive tasks involving temporal sequences. Facilitating the emergence of appropriate attractors during training is a difficult task that challenges neuroscientists even today. The authors alleviate this problem by first training RNNs on simple abstract tasks to allow tree-structured attractor templates to appear within the network, and then further training the networks on more complex tasks, such as a key-word spotting task which will effectively reuse the existing templates. This idea is similar to the emerging field of 'schemas' in network neuroscience.

On a technical standpoint, I am convinced that this work represents novel and significant advances in the field of biological RNNs, specifically in the subfield of schema formation. However, I have certain doubts on the neuroscientific elements of this work, especially since the paper is thoroughly cited with neuroscience literature. None of these concerns are a major detriment to my evaluation, and hence I recommend an accept for this work. However, I feel that the authors can considerably improve their discussion and neuroscientific arguments regarding the reusage of tree-structured templates. As such, I hope the authors can adequately address my concerns in the weaknesses section to further solidify this work.

**Strengths:**

The model and training methods have been extensively explained, which makes their objective clear. The underlying intuition that is being conveyed is straightforward and easy to understand, which is the direct consequence of a well-written introduction from pages 1 to 4. Subsequent sections retain the same quality of writing and scientific rigor.

**Weaknesses:**

As mentioned before, the key issues that I have about this work stems from neuroscientific plausibility. I do not have specific questions or requests regarding each point, and thus I would recommend the authors simply address this points by arguing for or against these statements.

1. It seems like in this work, the network needs to forget about the previous task in order to use an existing tree-structured template. This seems unrealistic for such a model, and seems to suggest a slot-based usage for such attractors.

2. There exists a lesser-known question in biological RNNs: suppose a network is tasked to remember 3 two-digit numbers (including 00) in sequence. The number of attractors required for this task would be $100^3 = 1,000,000$, which is exaggeratingly large for such a simple task that most humans are able to do. This deters the approach of using tree-structured attractors.

3. Discrete tree-structured attractors are also unable to account for continuous variables, such as color, intensity, speed, temperature etc. Yet, there is no simple generalization for this work to account for continuous variables.

4. Reusing a template also implies that the depth of the tree is fixed. While a tree can indeed grow deeper after subsequent training on more complex tasks, or become shorter due to lack of usage, it does not agree with the logical viewpoint that cognition is highly adaptive to sequence length at much shorter timescales compared to actually learning and developing tree-structures of modified depths.

5. Tree-structures may also lead to unintended behaviors if the wrong inputs are provided, especially if the inputs are not within the scope of the task. This leads to involuntary usage of these tree structures, thus presenting an inherent inflexibility in tree structures.


**Questions:**

See weaknesses.

**Limitations:**

I was not able to find explicit statements regarding limitations, although the typical shortcomings of a biological RNNs have been addressed as future research directions in lines 294-309. Points addressed in the weaknesses may possibly be added as limitations.

---

> ### Author Rebuttal · Authors · 2023-08-09
>
> Thanks for the valuable comments, which are very helpful for us to improve the paper. Below are our replies the comments point-by-point.
>
> On weaknesses:
>
> 1. Thank for raising this important issue. Shortly speaking, reusing an existing tree-structured template for a new task in our model does not need to forget about the previous task. In our model, we consider a recurrent network reserved to store tree-structure templates, a kind of slot-based memory. This is motivated by experimental findings which indicate that the brain recruits independent resources to store ordinal structures of temporal sequences disentangled from contents [1]. The advantages of this disentangled ordinal structure representation are that: 1) it saves the overall resource for representing a large number of temporal sequences sharing ordinal structures; 2) it allows the brain to reuse the shared ordinal structures, the so-called schema, to process different sequences flexibly. Once upon a temporal sequence arrives, the neural system rapidly binds the ordinal template with the contents. To process a new task, the neural system needs not to forget about the previous one; rather the neural system learns a new set of feedforward connections from the sensory cortex through dynamical binding. It has been explored previously that dynamic binding can be implemented in the brain through different means [2], including the fast Hebbian rule [3] and neural oscillation [4]. Notably, in the language domain, this schema idea is consistent with the minimalist program in linguistics proposed by Chomsky, which suggests that the brain stores many fundamental syntactic structures and is able to generate arbitrary complex syntactic trees via combining and reusing these primitive structures, referred to as merge, to realize fast language acquisition and appreciation [5].
>
> 2. The reviewer raised an interesting issue about the memory storage problem. Actually, disentangling ordinal structure from contents as a schema is a way for the brain to avoid the memory storage explosion problem. For a large number of sequences sharing the same ordinal structure, the brain only needs to store the ordinal template; while when processing a sequence with specific contents, the brain can quickly bind the template with the contents via dynamical binding.
>
> 3. We agree that our model can not account for continuous variables, but this is not a problem for the brain. A large volume of experimental studies has shown that the brain employs a different type of network model, called continuous attractor neural networks, to process continuous variables, such as head-direction [6] and spatial location [7]. This is not contradictory to our proposal of using tree-structured attractors to represent the ordinal structure of temporal sequences formed by discrete items.
>
> 4. Thank the reviewer for raising this important issue we have not addressed adequately in the original manuscript. Actually, our model has the capability to adapt to processing temporal sequences of varying lengths. First, for a sequence shorter than the template, our study has already demonstrated that our model can adapt to this, see Fig.3C-D in the manuscript, in which the network learned new sequences only covering part of the stored tree structure. Second, for a sequence longer than the template, our model can learn to combine shorter primitive templates to form a longer one, a strategy similar to the one proposed by Chomsky for language processing [5]. To demonstrate this idea, we conduct an additional experiment, in which the neural system learns to process sequences of depth 3 by combining two primitive templates of depth 2, see the new Fig.R1 in the reply pdf.
>
> 5. Thank the reviewer for pointing out this interesting issue which actually highlights an advantage of our model. In the conventional heteroclinic channel model, each node is an unstable saddle point, which can lead to unintended usage of the tree structure if the initial input is wrong. However, in our model, each node is a stable attractor, and to move to the next node, it needs the next input item to be correct in order to induce the transition (see lines 186-189 in the manuscript). As a result, our model displays the event-driven and evidence-accumulation behaviors (see Fig.4C-D in the manuscript). Overall, our model is rather robust to noises as it employs a sequence of attractors to represent the information.
>
> On limitations:
>
> We appreciate the reviewer's insightful comments and will add discussions about the limitations as suggested by the reviewer in the revised manuscript.
>
> References
>
> [1]	Dehaene et al, Neuron, 2015
>
> [2]	Engel Trends in cognitive sciences,2001
>
> [3]	Bittner Science,2017
>
> [4]	Klimesch et al,Neuroscience & Biobehavioral Reviews, 2010
>
> [5]	N. Chomsky, MIT Press, 1995
>
> [6]	Kim et al, Science, 2017
>
> [7]	Giocomo et al, Neuron, 2011

---

> > ### Comment · Reviewer_7Qop · 2023-08-12
> >
> > I thank the authors for the reply.
> >
> > > disentangling ordinal structure from contents...
> >
> > In general, I feel like the authors are defending the idea of disentangled representations in points 1 and 2, rather than the methodology in this paper. In this work, the authors retrain input and output weights in order to re-use tree structures. This is motivated by the idea of disentangled representations, which is a similar high-level concept, but they are fundamentally very different. For example, there is no mention of proposed mechanisms for "binding" in this context. Specifically, in the aforementioned task to remember 3 numbers, it is not possible to "bind" a number to an attractor simply by training a new set of output weights whose purpose is to decode 1 of 100 numbers.
> >
> > Quick update 5 minutes later: I am referring to training a network on a task to remember 3 arbitrary numbers, not 3 specific numbers such that it is possible to train input weights to bind them to attractors.
> >
> > > for a sequence longer than the template, our model can learn to combine shorter primitive templates to form a longer one
> >
> > I thank the authors for the additional results which has convinced me of their effectiveness.
> >
> > > highlights an advantage of our model
> >
> > What I mean is that an unintended input will cause dynamical systems to detract to unwanted stable attractors and stay there. I understand that the authors are trying to say that stable attractors are resistant to unintended inputs, but at this point I believe this discussion is too abstract and I will not consider the good or bad of this for my decision.
> >
> > I thank the authors again for the response and for the reasons above I will keep my current score, but I optionally invite the authors to respond to this for other reviewers and AC if there are any pressing points to be made (which I will also read).

---

> > > ### Author Response · Authors · 2023-08-13
> > >
> > > Thank you for your kind and insightful comments. Please let us explain further.
> > >
> > > Our emphasis on disentanglement stems from the fact that the learned structured attractors offer the advantage of facilitating the learning of new tasks with ease, along with the ability to effectively generalize across a wide range of sequential task classifications.
> > >
> > > Regarding point 1:
> > > For example, when learning a new 4-class phoneme sequence task, which includes sequence elements such as "pcl", "pau" and "dcl", and shares the same task structure with a 4-class synthetic sequence task (illustrated in Fig.1 of our manuscript), we can simply freeze the learned recurrent and readout connections while concentrating solely on learning new feedforward connections between phoneme inputs and the recurrent network, using backpropagation through time. As the synthetic task and phoneme task have different feedforward connections, the network can both perform the synthetic and phoneme tasks.
> > >
> > > Regarding point 2:
> > > Consider a simplified scenario in which we exclusively employ templates of depth 2, to address the challenge of a 100^3-class sequence classification task involves a two-fold process. Firstly, we learn a multitude of attractor templates independently, each with a length of 2, utilizing our methods. This compilation of templates serves as a reservoir of schemas. And this reservoir receives inputs and generates predictions through a linear layer as in Fig.R1 in the reply pdf, Secondly, we can only learn the input-output connections while preserving the recurrent connections in schemas. Distinct attractors from different schemas are able to represent sequence variables in a distributed manner. In an ideal scenario, 50 attractors from 50 schemas could collectively represent a formidable 2^50 variables. Similarly, attractor trajectories within each template also contributes to representing diverse sequence trajectories in a combinatorial manner. Consequently, the demand for a substantial 100^3 attractors is circumvented.
> > >
> > > Regarding point 5:
> > > In the case that an unintended input leading unwanted stable attractors, we agree that our model may fail. However, it is worth noting that such instances may indicate the presence of corrupted sequence inputs fraught with substantial noise. This could similarly lead to failure in other sequence models, including reservoir networks or heteroclinic channels. To address this challenge, a plausible solution may involve the implementation of a hierarchical recurrent network, where neurons in the high-level layer can integrate lengthy sequence elements and help to rectify unintended behaviors.
> > >
> > > We thank the reviewers for their valuable criticism, which we intend to incorporate into our future studies.

---

> > > > ### Comment · Reviewer_7Qop · 2023-08-13
> > > >
> > > > I thank the authors for the quick reply.
> > > >
> > > > I would like to reiterate that this work is important for biological RNN audiences, and this discussion here does not critique the experiments already reported in the paper -- I feel they are sufficient and rigorous.
> > > >
> > > > As for the current reply, I acknowledge that viewing the brain as a binary system like a computer operating using bits is a valid mathematical solution to the problem.

---

> > > > > ### Author Response · Authors · 2023-08-14
> > > > >
> > > > > Thanks again to the reviewer for the affirmation and valuable criticism.

---

### Author Rebuttal · Authors · 2023-08-10

We appreciate the valuable comments from all reviewers, which are very helpful for us to improve the work. We have addressed all concerns of the reviewers point-by-point.

Attached please find the supplementary figures to answer the concerns of reviewers.

---

### Decision · Program_Chairs · 2023-09-21

**Decision:**

Accept (poster)

**Comment:**

The paper investigates how recurrent networks can rapidly learn ordinal information, via pertaining a recurrent RNN such that it learns tree-structured templates that can be rapidly bound to new items.

Strengths:

-Clear and complete explanation of methods and objective

-The problem is important

-The proposed method achieves good performance and behaviour consistent with neuroscientific studies; it also provides a compelling intuition for transfer learning abilities

Weaknesses:

-The paper would benefit from additional discussion of related sequence models in the brain

-The neural plausibility is not clear

-How the brain could achieve training on diverse temporal intervals is unclear

-The evaluation focuses on small sequences and the scalability of the method is unclear